# Diffusion with Forward Models: Solving Stochastic Inverse Problems Without Direct Supervision

**Ayush Tewari**[1][*]     **Tianwei Yin**[1][*]     **George Cazenavette**[1]     **Semon Rezchikov**[4]
**Joshua B. Tenenbaum**[1,2,3]     **Frédo Durand**[1]     **William T. Freeman**[1]     **Vincent Sitzmann**[1]

[1]MIT CSAIL     [2]MIT BCS     [3]MIT CBMM     [4]Princeton IAS

## Abstract

Denoising diffusion models have emerged as a powerful class of generative models capable of capturing the distributions of complex, real-world signals. However, current approaches can only model distributions for which training samples are directly accessible, which is not the case in many real-world tasks. In inverse graphics, for instance, we seek to sample from a distribution over 3D scenes consistent with an image but do not have access to ground-truth 3D scenes, only 2D images. We present a new class of conditional denoising diffusion probabilistic models that learn to sample from distributions of signals that are never observed directly, but instead are only measured through a known differentiable forward model that generates partial observations of the unknown signal. To accomplish this, we directly integrate the forward model into the denoising process. At test time, our approach enables us to sample from the distribution over underlying signals consistent with some partial observation. We demonstrate the efficacy of our approach on three challenging computer vision tasks. For instance, in inverse graphics, we demonstrate that our model in combination with a 3D-structured conditioning method enables us to directly sample from the distribution of 3D scenes consistent with a single 2D input image.

## 1   Introduction

Consider the problem of reconstructing a 3D scene from a single picture. Since much of the 3D scene is unobserved, there are an infinite number of 3D scenes that could have produced the image, due to the 3D-to-2D projection, occlusion, and limited field-of-view that leaves a large part of the 3D scene unobserved. Given the ill-posedness of this problem, it is desirable for a reconstruction algorithm to be able to sample from the distribution over all plausible 3D scenes that are consistent with the 2D image, generating unseen parts in plausible manners. Previous data-completion methods, such as in-painting in 2D images, are trained on large sets of ground-truth output images along with their incomplete (input) counterparts. Such techniques do not easily extend to 3D scene completion, since curating a large dataset of ground-truth 3D scene representations is very challenging.

This 3D scene completion problem, known as inverse graphics, is just one instance of a broad class of problems often referred to as *Stochastic Inverse Problems*, which arise across scientific disciplines whenever we capture partial observations of the world through a sensor. In this paper, we introduce a diffusion-based framework that can tackle this problem class, enabling us to sample from a distribution of signals that are consistent with a set of partial observations that are generated from the signal by a non-invertible, generally nonlinear, forward model. For instance, in inverse graphics, we learn to sample 3D scenes given an image, yet never observe paired observations of images and 3D scenes at training time, nor observe 3D scenes directly.

---

[*] Equal Contribution. Project page: diffusion-with-forward-models.github.io

37th Conference on Neural Information Processing Systems (NeurIPS 2023).

While progress in deep learning for generative modeling has been impressive, this problem remains unsolved. In particular, variational autoencoders and conditional neural processes are natural approaches but have empirically fallen short of modeling the multi-modal distributions required in, for instance, inverse graphics. They have so far been limited to simple datasets. Emerging diffusion models [1], in contrast, enable sampling from highly complex conditional distributions but require samples from the output distribution that is to be modeled for training, e.g. full 3D models. Some recent work in inverse graphics has resorted to a two-stage approach, where one first reconstructs a large dataset of 3D scenes to then train an image-conditional diffusion model to sample from the conditional distribution over these scenes [2, 3]. To avoid a two-stage approach, another recent line of work trains a conditional diffusion model to sample from the distribution over novel views of a scene, only requiring image observations at training time [4, 5]. However, such methods do *not* model the distribution over 3D scenes directly and therefore cannot sample from the distribution over 3D scenes consistent with an image observation. Thus, a multi-view consistent 3D scene can only be obtained in a costly post-processing stage [6]. A notable exception is the recently proposed RenderDiffusion [7], demonstrating that it is possible to train an unconditional diffusion model over 3D scenes from observing only monocular images. While one can perform conditional sampling even with unconditional models, they are fundamentally limited to simple distributions, in this case, single objects in canonical orientations.

Our core contribution is a novel approach for integrating any differentiable forward model that describes how partial observations are obtained from signals, such as 2D image observations and 3D scenes, with conditional denoising diffusion models. By sampling an observation from our model, we jointly sample the signal that gave rise to that observation. Our approach has a number of advantages that make it highly attractive for solving complex Stochastic Inverse Problems. First, our model is trained end-to-end and does away with two-stage approaches that first require reconstruction of a large dataset of signals. Second, our model directly yields diverse samples of the signal of interest. For instance, in the inverse graphics setting, our model directly yields highly diverse samples of 3D scenes consistent with an observation that can then be rendered from novel views with guaranteed multi-view consistency. Finally, our model naturally leverages domain knowledge in the form of known forward models, such as differentiable rendering, with all guarantees that such forward models provide. We validate our approach on three challenging computer vision tasks: inverse graphics (the focus of this paper), as well as single-image motion prediction and GAN inversion.

In summary, we make the following contributions:

1. We propose a new method that integrates differentiable forward models with conditional diffusion models, replacing prior two-step approaches with a conditional generative model trained end-to-end.

2. We apply our framework to build the first conditional diffusion model that learns to sample from the distribution of 3D scenes trained only on 2D images. In contrast to prior work, we *directly* learn image-conditional 3D radiance field generation, instead of sampling from the distribution of novel views conditioned on a context view. Our treatment of inverse graphics exceeds a mere application of the proposed framework, contributing a novel, 3D-structured denoising step that leverages differentiable rendering both for conditioning and for the differentiable forward model.

3. We formally prove that under natural assumptions, as the number of observations of each signal in the training set goes to infinity, the proposed model maximizes not only the likelihood of observations, but also the likelihood of the unobserved signals.

4. We demonstrate the efficacy of our model for two more downstream tasks with structured forward models: single-image motion prediction, where the forward model is a warping operation, and GAN inversion, where the forward model is a pretrained StyleGAN [8] generator.

## 2 Method

Consider observations $(\mathbf{O}_j^i, \phi_j^i)$ that are generated from underlying signals $\mathbf{S}_j$ according to a known forward model `forward()`, i.e., $\mathbf{O}_j^i = \texttt{forward}(\mathbf{S}_j, \phi_j^i)$, where $\phi_j^i$ are parameters of the forward model corresponding to observation $\mathbf{O}_j^i$. Each observation can be *partial*. Specifically, given a *single* observation, there is an infinite number of signals that could have generated this observation. However, we assume that given a hypothetical set of *all possible* observations, the signal is fully

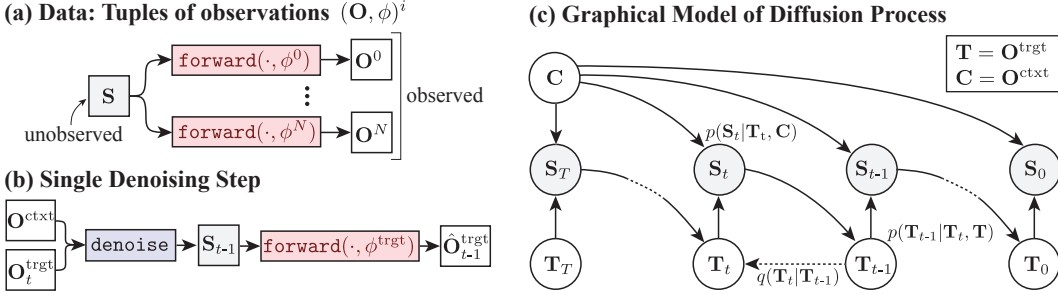

**(a) Data: Tuples of observations** $(\mathbf{O}, \phi)^i$

observed

unobserved

**(b) Single Denoising Step**

**(c) Graphical Model of Diffusion Process**

Figure 1: **Overview of our proposed method.** **(a)** We assume a dataset of tuples of observations $(\mathbf{O}, \phi)^i$, generated from *unobserved* signals $\mathbf{S}$ via a differentiable forward model. **(b)** We propose to integrate the forward model directly into the denoising step of a diffusion model: given a pair of observations of the same signal, we designate context $\mathbf{O}^{\text{ctxt}}$ and target $\mathbf{O}^{\text{trgt}}$. We add noise to $\mathbf{O}^{\text{trgt}}$, then feed $(\mathbf{O}^{\text{ctxt}}, \phi^{\text{ctxt}}, \mathbf{O}^{\text{trgt}}_t, \phi^{\text{trgt}})$ to a neural network `denoise` to estimate the signal $\mathbf{S}_{t\text{-}1}$. We then apply the forward model to obtain an estimate of the clean target observation, $\hat{\mathbf{O}}^{\text{trgt}}_{t\text{-}1}$. **(c)** The graphical model of the diffusion process.

determined. In the case of inverse graphics, $\mathbf{O}^i_j$ are image observations of 3D scenes $\mathbf{S}_j$ and $\phi^i_j$ are the camera parameters, where we index scenes with $j$ and observations of the $j$-th scene via $i$. `forward()` is the rendering function. Note that if we were to capture *every possible image* of a 3D scene, the 3D scene is uniquely determined, but given a *single* image, there are an infinite number of 3D scenes that could have generated that image, both due to the fact that rendering is a projection from 3D and 2D, and due to the fact that a single image only constrains the visible part of the 3D scene. We will drop the subscript $j$ in the following, and leave it implied that we always consider *many* observations generated from *many* signals. Fig. 1 provides an illustration of the data.

We are now interested in training a model that, at test time, allows us to sample from the distribution of signals that are consistent with a previously unseen observation $\mathbf{O}$. Formally, we aim to model the conditional distribution $p(\mathbf{S}|\mathbf{O}, \phi)$. We make the following assumptions:

- We have access to a differentiable implementation of `forward()`.
- We have access to a large dataset of observations and corresponding parameters of the forward model, $\{(\mathbf{O}^i, \phi^i)\}^N_i$.
- In our training set, we have access to *several* observations per signal.

Crucially, we do *not* assume that we have direct access to the underlying signal that gave rise to a particular observation, i.e., we do *not* assume access to tuples of $(\mathbf{O}, \phi, \mathbf{S})$. Further, we also do *not* assume that we have access to any prior distribution over the signal of interest, i.e., we never observe a dataset of signals of the form $\{\mathbf{S}^j\}_j$, and thus cannot train a generative model to sample from an unconditional distribution over signals.

Recent advances in deep-learning-based generative modeling have seen the emergence of denoising diffusion models as powerful generative models that can be trained to generate highly diverse samples from complex, multi-modal distributions. We are thus motivated to leverage denoising diffusion probabilistic models to model $p(\mathbf{S}|\mathbf{O}, \phi)$. However, existing approaches cannot be trained if we do not have access to signals $\mathbf{S}$. In the following, we give background on denoising diffusion models and discuss the limitation.

## 2.1 Background: Denoising Diffusion Probabilistic Models and their Limitation

Denoising diffusion probablistic models are a class of generative models that learn to sample from a distribution by learning to iteratively denoise samples. Consider the problem of modeling the distribution $p_\theta(\mathbf{x})$ over samples $\mathbf{x}$. A forward Markovian process $q(\mathbf{x}_{0:T})$ adds noise to the data as

$$q(\mathbf{x}_t \mid \mathbf{x}_{t-1}) = \mathcal{N}(\mathbf{x}_t; \sqrt{1 - \beta_t}\mathbf{x}_{t-1}, \beta_t\mathbf{I}). \tag{1}$$

Here, $\beta_t$, $t \in 1 \ldots T$ are the hyperparameters that control the variance schedule. A denoising diffusion model learns the reverse process, where samples from a distribution $p(x_T) = \mathcal{N}(\mathbf{0}, \mathbf{I})$ are transformed incrementally into the data manifold as $p_\theta(\mathbf{x}_{0:T}) = p(x_T) \prod_{t=1}^T p_\theta(\mathbf{x}_{t-1} \mid \mathbf{x}_t)$, where

$$p_\theta(\mathbf{x}_{t-1} \mid \mathbf{x}_t) = \mathcal{N}(\mathbf{x}_{t-1}; \mu(\mathbf{x}_t, t), \Sigma(\mathbf{x}_t, t)). \tag{2}$$

A neural network `denoise`$_\theta()$ with learnable parameters $\theta$ learns to reverse the diffusion process. It is also possible to model conditional distributions $p_\theta(\mathbf{x}_{0:T} \mid \mathbf{c})$, where the output is computed as

`denoise`$_\theta(\mathbf{x}_t, t, \mathbf{c})$. The forward process does not change in this case; in practice, we merely add the conditional signal as input to the denoising model.

Unfortunately, we cannot train existing denoising diffusion models to sample from $p(\mathbf{S} \mid \mathbf{O}, \phi)$, or, in fact, even from an unconditional distribution $p(\mathbf{S})$. This would require computation of the Markovian forward process in Eq. 1. However, recall that we do not have access to any signals $\{\mathbf{S}^j\}_j$ - we thus can not add any noise to any signals to then train a denoising neural network. In other words, since no $\mathbf{S}$ is directly observed, we *cannot* compute $q(\mathbf{S}_t \mid \mathbf{S}_{t-1})$.

## 2.2 Integrating Denoising Diffusion with Differentiable Forward Models

We now introduce a class of denoising diffusion models that we train to directly model the distribution $p(\mathbf{S} \mid \mathbf{O}^{\text{ctxt}}; \phi^{\text{ctxt}})$ over signals $\mathbf{S}$ given an observation $(\mathbf{O}^{\text{ctxt}}, \phi^{\text{ctxt}})$. Our key contribution is to directly integrate the differentiable forward model `forward()` into the iterative conditional denoising process. This enables us to add noise to and denoise the observations, while nevertheless sampling the underlying signal that generates that observation.

Our model is trained on pairs of "context" and "target" observations of the same signal, denoted as $\mathbf{O}^{\text{ctxt}}$ and $\mathbf{O}^{\text{trgt}}$. As in conventional diffusion models, for the forward process, we have $q(\mathbf{O}_t^{\text{trgt}} \mid \mathbf{O}_{t-1}^{\text{trgt}}) = \mathcal{N}(\mathbf{O}_t^{\text{trgt}}; \sqrt{1-\beta_t}\mathbf{O}_{t-1}^{\text{trgt}}, \beta_t\mathbf{I})$. In the reverse process, we similarly denoise $\mathbf{O}^{\text{trgt}}$ conditional on $\mathbf{O}^{\text{ctxt}}$:

$$p_\theta(\mathbf{O}_{0:T}^{\text{trgt}} \mid \mathbf{O}^{\text{ctxt}}; \phi^{\text{ctxt}}, \phi^{\text{trgt}}) = p(\mathbf{O}_T^{\text{trgt}}) \prod_{t=0}^{T} p_\theta(\mathbf{O}_{t-1}^{\text{trgt}} \mid \mathbf{O}_t^{\text{trgt}}, \mathbf{O}^{\text{ctxt}}; \phi^{\text{ctxt}}, \phi^{\text{trgt}}), \tag{3}$$

However, unlike conventional diffusion models, we implement $p_\theta(\mathbf{O}_{t-1}^{\text{trgt}} \mid \mathbf{O}_t^{\text{trgt}}, \mathbf{O}^{\text{ctxt}}; \phi^{\text{ctxt}}, \phi^{\text{trgt}})$ by first predicting an estimate of the underlying signal $\mathbf{S}_{t-1}$ and then mapping it to an estimate of the denoised observations via the differentiable `forward`:

$$\mathbf{S}_{t-1} = \texttt{denoise}_\theta(\mathbf{O}^{\text{ctxt}}, \mathbf{O}_t^{\text{trgt}}; t, \phi^{\text{ctxt}}, \phi^{\text{trgt}}), \tag{4}$$

$$\hat{\mathbf{O}}_{t-1}^{\text{trgt}} = \texttt{forward}(\mathbf{S}_{t-1}, \phi^{\text{trgt}}) \tag{5}$$

$$\mathbf{O}_{t-1}^{\text{trgt}} \sim \mathcal{N}(\mathbf{O}_{t-1}^{\text{trgt}}; C_{t-1}\hat{\mathbf{O}}_{t-1}^{\text{trgt}}, \hat{\beta}_{t-1}\mathbf{I}) \tag{6}$$

Here, $\hat{\mathbf{O}}_{t-1}^{\text{trgt}}$ is an estimate of the *clean* observation, and the constants $C_{t-1}$ and $\hat{\beta}_{t-1}$ are chosen to match the total noise added by the forward process at time $t$-1. See Fig. 1 for an overview. At test time, a signal is sampled by iterating Eq. 4, 5, and 6 starting with $p(\mathbf{O}_{t=T}^{\text{trgt}}) \sim \mathcal{N}(\mathbf{0}, \mathbf{I})$. Importantly, our models define a generative model over the underlying signal via Eq. 4:

$$p_{\theta, \phi^{\text{trgt}}}(\mathbf{S}_{0:T} \mid \mathbf{O}^{\text{ctxt}}; \phi^{\text{ctxt}}) = \prod_{t=1}^{T} p_\theta(\mathbf{S}_{t-1} \mid \mathbf{O}_t^{\text{trgt}}, \mathbf{O}^{\text{ctxt}}; \phi^{\text{ctxt}}, \phi^{\text{trgt}}). \tag{7}$$

We will suppress the subscript in the notation, and refer to this distribution as $p(\mathbf{S}_{0:T} \mid \mathbf{O}^{\text{ctxt}}; \phi^{\text{ctxt}})$ for brevity from now.

**Loss Function.** We train to minimize the following two loss terms:

$$\mathcal{L}_\theta^{\text{trgt}} = \mathbb{E}_{\mathbf{O}^{\text{ctxt}}, \mathbf{O}^{\text{trgt}}, \phi^{\text{ctxt}}, \phi^{\text{trgt}}, t}\left[\|\mathbf{O}^{\text{trgt}} - \underbrace{\texttt{forward}(\texttt{denoise}_\theta(\mathbf{O}^{\text{ctxt}}, \mathbf{O}_t^{\text{trgt}}; t, \phi^{\text{ctxt}}, \phi^{\text{trgt}}), \phi^{\text{trgt}})}_{=\hat{\mathbf{O}}_{t-1}^{\text{trgt}}}\|^2\right],$$
$$\tag{8}$$

$$\mathcal{L}_\theta^{\text{novel}} = \mathbb{E}_{\mathbf{O}^{\text{ctxt}}, \mathbf{O}^{\text{novel}}, \phi^{\text{ctxt}}, \phi^{\text{trgt}}, \phi^{\text{novel}}, t}\left[\|\mathbf{O}^{\text{novel}} - \underbrace{\texttt{forward}(\texttt{denoise}_\theta(\mathbf{O}^{\text{ctxt}}, \mathbf{O}_t^{\text{trgt}}; t, \phi^{\text{ctxt}}, \phi^{\text{trgt}}), \phi^{\text{novel}})}_{=\hat{\mathbf{O}}_{t-1}^{\text{novel}}}\|^2\right].$$
$$\tag{9}$$

Here, we compute the estimate of the observation from the target, as well as a separate, novel forward model parameter $\phi^{\text{novel}}$. In the supplemental document, we show that these losses approximate a total observation loss, maximizing the likelihood of all possible observations of the signal $\mathbf{S}$.

**Characterizing the Conditional Distribution Over Signals.** Due to the complexity of the reverse process, it may not be clear that the learned distribution over signals will agree with the true distribution, even in the limit of infinite data. However, this model will indeed asymptotically learn the true conditional distribution over signals, as we formally prove in the supplement:

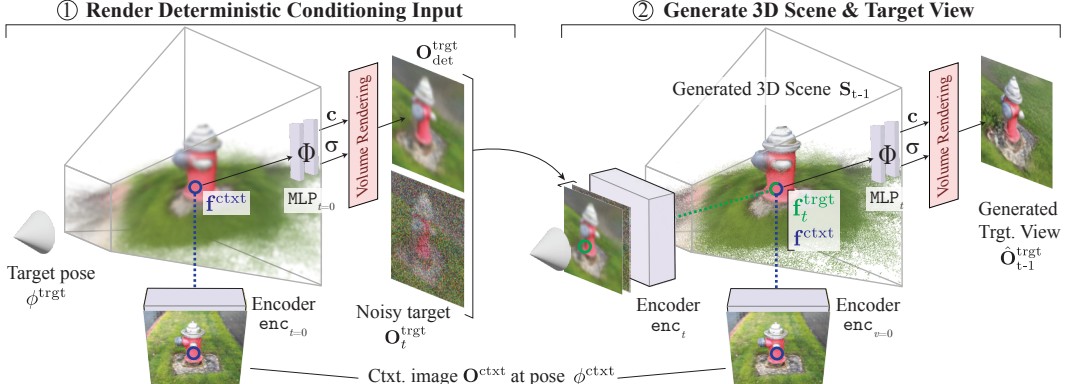

Figure 2: **Overview of 3D Generative Modeling.** We build a 3D-structured `denoise` operator on top of pixelNeRF [9] that learns to sample from the distribution of 3D scenes from image observations only. Given a context image $\mathbf{O}^{\text{ctxt}}$ with camera pose $\phi^{\text{ctxt}}$, we pick a target pose $\phi^{\text{trgt}}$. We render out a deterministic estimate of the depth, RGB, and features of the target view $\mathbf{O}^{\text{trgt}}_{\text{det}}$ using pixel-aligned features $\mathbf{f}^{\text{ctxt}}$ extracted from the context view with encoder $\text{enc}_{t=0}$ (left, only RGB shown here). To generate a 3D scene, we concatenate the deterministic estimate with noise $\mathbf{O}^{\text{trgt}}_t$, and extract features $\mathbf{f}^{\text{trgt}}_t$ for the *target* view with $\text{enc}_t$. $\mathbf{f}^{\text{trgt}}_t$ and $\mathbf{f}^{\text{ctxt}}$ now jointly parameterize the radiance field of the generated scene $\mathbf{S}_{t-1}$, and we may render an estimate of the clean target view $\hat{\mathbf{O}}^{\text{trgt}}_{t-1}$. The model is trained end-to-end via a re-rendering loss.

**Proposition 1.** Suppose that any signal $\mathbf{S}$ can be reconstructed from the set of all *all possible* observations of $\mathbf{S}$. Under this assumption, if in the limit as the number of known observations per signal goes to infinity, there are parameters $\theta$ such that $\mathcal{L}^{\text{trgt}}_{\theta} + \mathcal{L}^{\text{novel}}$ is minimized, then the conditional probability distribution over signals discovered by our model $p(\mathbf{S} \mid \mathbf{O}^{\text{ctxt}}; \phi^{\text{ctxt}})$ agrees with the true distribution $p^{\text{true}}(\mathbf{S} \mid \mathbf{O}^{\text{ctxt}}; \phi^{\text{ctxt}})$.

The proof follows by showing that our losses implicitly minimize a diffusion model loss over *total observations*, which are collections of all possible observations of our signal. As such, when the observations suffice to completely reconstruct the signal, the correctness of the estimated distribution over total observations forces the estimated distribution over signals to be correct, as well.

## 3 Prior Work on Latent Variable Models for Inverse Problems

Variational Autoencoders [10, 11], normalizing flows [12], conditional [13] and attentive neural processes [14] are latent-variable models that can be combined with forward models to learn to sample from the distribution of unobserved signals from observations [15, 16]. However, they empirically fall short of accurately modeling complex signal distributions - in inverse graphics, for instance, such models have so far been limited to synthetic 3D scenes. Generative Adversarial Networks can be trained with differentiable forward models in-the-loop, and have yielded impressive results in unconditional generative modeling of unobserved signals [17–19]. Similarly, in concurrent work, diffusion models have been leveraged for unconditional generative modeling through differentiable forward models [2, 7, 20]. However, unconditional models are limited to tight distributions, and no conditional generative modeling of similar quality has been demonstrated. Diffusion models trained directly on signals have been effectively applied to diverse inverse problems such as super-resolution [21–25], inpainting [21, 23–26], and medical imaging [27]. These works utilize the learned prior of the data distribution to recover the latent signal through a "plug and play" approach [28–30], integrating the diffusion model with a forward measurement process according to Bayes' rule. These approaches are versatile and can easily adapt to new inverse problems without retraining. However, unlike our models, they rely on direct supervision over the signals in the form of large datasets.

## 4 Applications

We now apply our framework to three stochastic inverse problems. We focus on applications in computer vision, where we tackle the problems of inverse graphics, single-image motion prediction, and GAN inversion. For each application, we give a detailed description of the forward model, the dataset and baselines, as well as a brief description of prior work.

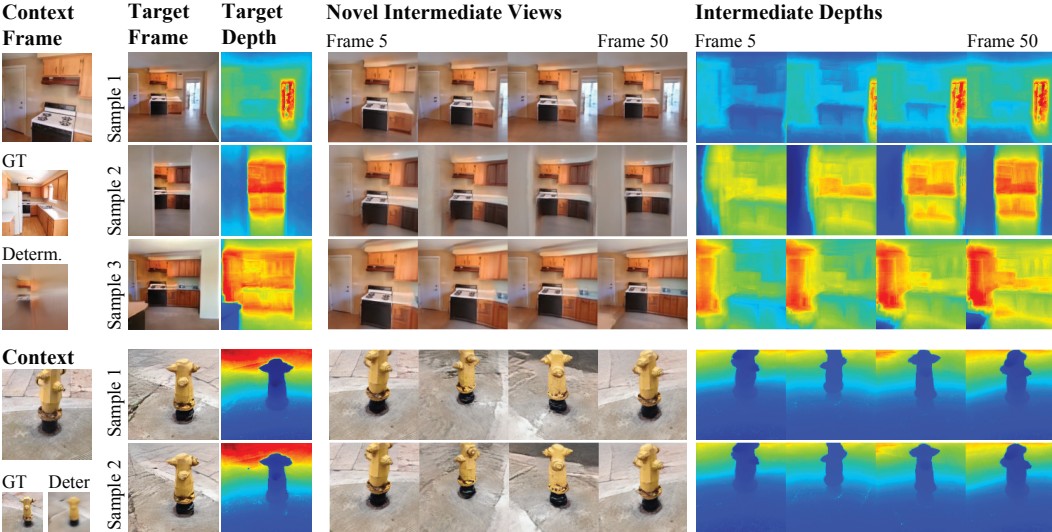

| Context Frame | Target Frame | Target Depth | Novel Intermediate Views | Intermediate Depths |
|---|---|---|---|---|

Figure 3: **Sample Diversity.** We illustrate different 3D scenes sampled from the same context image for RealEstate10k and Co3D datasets. Unlike deterministic methods like pixelNeRF [9], our method generates diverse and distinct 3D scenes that all align with the context image. Co3D results are generated using autoregressive sampling, where a 360 degree trajectory can be generated by iteratively sampling target images. Note the photorealism and diversity of the generated structures for the indoor scene, such as doors and cabinets. Also note the high-fidelity geometry of the occluded parts of the hydrant and the diverse background appearance.

## 4.1 Inverse Graphics

We seek to learn a model that, given a single image of a 3D scene enables us to sample from the distribution over 3D scenes that are consistent with the observation. We expect that 3D regions visible in the image are reconstructed faithfully, while unobserved parts are generated plausibly. Every time we sample, we expect a *different* plausible 3D generation. Signals $\mathbf{S}$ are 3D scenes, and observations are 2D images $\mathbf{O}$ and their camera parameters $\phi$. At training time, we assume that we have access to at least two image observations and their camera parameters per scene, such that we can assemble tuples of $(\mathbf{O}^{\text{ctxt}}, \phi^{\text{ctxt}}, \mathbf{O}^{\text{trgt}}, \phi^{\text{trgt}})$, with 2D images $\mathbf{O}^{\text{ctxt}}, \mathbf{O}^{\text{trgt}}$, and camera parameters $\phi^{\text{ctxt}}, \phi^{\text{trgt}}$.

**Scope.** We note that our treatment of inverse graphics exceeds a mere application of the presented framework. In particular, we not only integrate the differentiable rendering `forward` function, but further propose a novel 3D-structured `denoise` function. Here, we enable state-of-the-art conditional generation of complex, real-world 3D scenes.

**Related Work.** Few-shot reconstruction of 3D scene representations via differentiable rendering was pioneered by deterministic methods [9, 31, 32, 32–41] that blur regions of the 3D scene unobserved in the context observations. Probabilistic methods have been proposed that can sample from the distribution of novel views trained only on images [4, 5, 42–45]. While results are impressive, these methods do not allow sampling from the distribution of *3D scenes*, but only from the distribution of *novel views*. Generations are not multi-view consistent. Obtaining a 3D scene requires costly post-processing via score distillation [6]. Several approaches [2, 3] use a two-stage design: they first reconstruct a dataset of 3D scenes, and then train a 3D diffusion model. However, pre-computing large 3D datasets is expensive. Further, to obtain high-quality results, dense observations are required per scene. RenderDiffusion [7] and HoloDiffusion [20] integrate differentiable forward rendering with an unconditional diffusion model, enabling unconditional sampling of simple, single-object scenes. Similar to us, RenderDiffusion performs denoising in the image space, while HoloDiffusion uses a 3D denoising architecture. Other methods use priors learned by text-conditioned image diffusion models to optimize 3D scenes [46–48]. Here, the generative model does not have explicit knowledge about the 3D information of scenes. These methods often suffer from geometric artifacts.

**Structure of S and forward model `render`.** We can afford only an abridged discussion here - please see the supplement for a more detailed description. We use NeRF [49] as the parameterization of 3D scenes, such that $\mathbf{S}$ is a function that maps a 3D coordinate $\mathbf{p}$ to a color $\mathbf{c}$ and density $\sigma$ as $\mathbf{S}(\mathbf{p}) = (\sigma, \mathbf{c})$. We require a *generalizable* NeRF that is predicted in a feed-forward pass by an encoder that takes a set of $M$ context images and corresponding camera poses $\{(\mathbf{O}_i, \phi_i)\}_i^M$ as input.

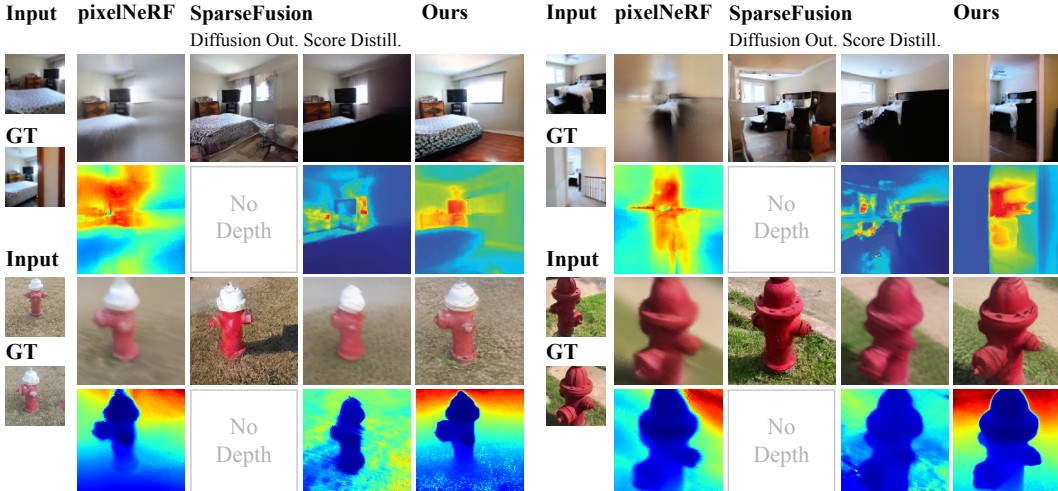

Figure 4: **Qualitative Comparison for Inverse Graphics application.** We benchmark with SparseFusion [5] and the deterministic pixelNeRF [9]. SparseFusion samples 2D novel views conditioned on a deterministic rendering (Diffusion Out.), and generates multi-view consistent 3D scenes only after Score Distillation. Our method consistently generates higher-quality scenes, while directly sampling 3D scenes.

We base our model on pixelNeRF [9]. pixelNeRF first extracts image features $\{\mathbf{F}_i\}_i$ from each context observation via an encoder enc as $\mathbf{F}_i = \texttt{enc}(\mathbf{O}_i)$. Given a 3D point $\mathbf{p}$, it obtains its pixel coordinates in each context view via $\mathbf{p}_i^{\text{pix}} = \pi(\mathbf{p}, \phi_i)$ via the projection operator $\pi$, and recovers a corresponding feature as $\mathbf{f}_i = \mathbf{F}_i(\mathbf{p}_i^{\text{pix}})$ by sampling the feature map at pixel coordinate $\mathbf{p}_i^{\text{pix}}$. It then parameterizes $\mathbf{S}$ via an MLP as:

$$\mathbf{S}(\mathbf{p}) = (\sigma(\mathbf{p}), \mathbf{c}(\mathbf{p})) = \texttt{MLP}(\{(\mathbf{f}_i \oplus \mathbf{p}_i\}_i^M), \tag{10}$$

where $\oplus$ is concatenation and $\mathbf{p}_i$ is the 3D point $\mathbf{p}$ transformed into the camera coordinates of observation $i$. The number of context images $M$ is flexible, and we may condition $\mathbf{S}$ on a single or several observations. It will be convenient to refer to a pixelNeRF that is reconstructed from context and target observations $(\mathbf{O}^{\text{ctxt}}, \phi^{\text{ctxt}})$ and $(\mathbf{O}^{\text{trgt}}, \phi^{\text{trgt}})$ as

$$\mathbf{S}(\cdot \mid \texttt{enc}(\mathbf{O}^{\text{ctxt}}), \texttt{enc}(\mathbf{O}^{\text{trgt}})), \tag{11}$$

where we make the pixelNeRF encoder enc explicit and drop the poses $\phi^{\text{trgt}}$ and $\phi^{\text{ctxt}}$. We leverage differentiable volume rendering [49] as forward model, such that

$$\mathbf{O} = \texttt{render}(\mathbf{S}, \phi), \tag{12}$$

where $\mathbf{S}$ is rendered from a camera with parameters $\phi$.

**Implementation of** `denoise`**.** Fig. 2 gives an overview of the denoising procedure. Following our framework, we obtain the denoised target observation $\hat{\mathbf{O}}_{t\text{-}1}^{\text{trgt}}$ as:

$$\hat{\mathbf{O}}_{t\text{-}1}^{\text{trgt}} = \texttt{render}(\mathbf{S}_{t\text{-}1}, \phi^{\text{trgt}}), \quad \text{where} \tag{13}$$
$$\mathbf{S}_{t\text{-}1} = \mathbf{S}(\cdot \mid \texttt{enc}_{t=0}(\mathbf{O}^{\text{ctxt}}), \texttt{enc}_t(\mathbf{O}_t^{\text{trgt}})), \tag{14}$$

where the image encoder $\texttt{enc}_t$ is now conditioned on the timestep $t$. In other words, we will generate a target view $\hat{\mathbf{O}}_{t\text{-}1}^{\text{trgt}}$ by rendering the pixelNeRF conditioned on the context and noisy target observations. However, feeding the noisy $\mathbf{O}_t^{\text{trgt}}$ directly to pixelNeRF is insufficient. This is because the pixel-aligned features $\texttt{enc}_t(\mathbf{O})$ are obtained from each view separately - thus, the features generated by $\texttt{enc}_t(\mathbf{O}_t^{\text{trgt}})$ will be uninformative. To successfully generate a 3D scene, we have to augment the $\mathbf{O}_t^{\text{trgt}}$ with information from the context view. We propose to generate conditioning information for $\mathbf{O}_t^{\text{trgt}}$ by rendering a *deterministic estimate* $\mathbf{O}_{\text{det}}^{\text{trgt}} = \texttt{render}\left(\mathbf{S}(\cdot \mid \texttt{enc}_{t=0}(\mathbf{O}^{\text{ctxt}})), \phi^{\text{trgt}}\right)$. I.e., we condition pixelNeRF only on the context view, and render an estimate of the target view via volume rendering. However, in the extreme case of a completely uncertain target view, this results in a completely blurry image. We thus propose to additionally render high-dimensional features. Recall that any 3D point $\mathbf{p}$, we have $(\sigma(\mathbf{p}), \mathbf{c}(\mathbf{p})) = \texttt{MLP}_t(\mathbf{p})$. We modify $\texttt{MLP}_t$ to also output a high-dimensional feature and

**Single-Image Motion Prediction**

Input    Motion Field Samples    Determ.

**Probabilistic GAN Inversion**

Patch    Samples from GAN inversion    Determ.

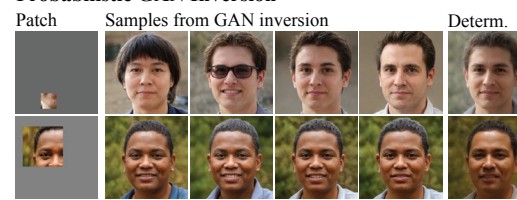

Figure 5: Qualitative Results for Single-Image Motion Prediction (left) and GAN Inversion (right).

render a deterministic feature map to augment $\mathbf{O}_t^{\text{trgt}}$ (only RGB shown in figure). We generate the final 3D scene as $\mathbf{S}_{t\text{-}1} = \mathbf{S}(\cdot \mid \text{enc}_{t=0}(\mathbf{O}^{\text{ctxt}}), \text{enc}_t(\mathbf{O}_{\text{det}}^{\text{trgt}} \oplus \mathbf{O}_t^{\text{trgt}}))$. The final denoised target view is then obtained according to the rendering Eq. 13 above.

**Loss and Training.** Our loss consists of simple least-squares terms on re-rendered views, identical to the general loss terms presented in Eqs. 8 and 9, in addition to regularizers that penalize degenerate 3D scenes. We discuss these regularizers, as well as training details, in the supplement.

### 4.1.1 Results

**Datasets** We evaluate on two challenging real-world datasets. We use Co3D hydrants [50] to evaluate our method on object-centric scenes. For scene-level 3D synthesis, we use the challenging RealEstate10k dataset [51], consisting of indoor and outdoor videos of scenes.

**Baselines** We compare our approach with state-of-the-art approaches in deterministic and probabilistic 3D scene completion. We use pixelNeRF as the representative method for deterministic methods that takes a single image as input and deterministically reconstructs a 3D scene. Our method is the first to probabilistically reconstruct 3D scenes in an end-to-end manner. Regardless, we compare with the concurrent SparseFusion [52] that learns an image-space generative model over novel views of a 3D scene. Score distillation of this generative model is required every time we want to obtain a multi-view consistent 3D scene, which is costly.

**Qualitative Results.** In Fig. 3, we show multiple samples of 3D scenes sampled from a monocular image. For the indoor scenes of RealEstate10k, there are large regions of uncertainty. We can sample from the distribution of valid 3D scenes, resulting in significantly different 3D scenes with plausible geometry and colors. The objects are faithfully reconstructed for the object-centric Co3D scenes, and the uncertainty in the scene is captured. We can sample larger 3D scenes and render longer trajectories by autoregressive sampling, i.e., we treat intermediate diffused images as additional context observations to sample another target observation. The Co3D results in Fig. 3 were generated autoregressively for a complete 360 degrees trajectory. In Fig. 4, we compare our results with pixelNeRF [9] and SparseFusion [5]. pixelNeRF is a deterministic method and thus leads to very blurry results in uncertain regions. SparseFusion reconstructs scenes by score-distillation over a 2D generative model. This optimization is very expensive, and does not lead to natural-looking results.

**Quantitative Results.** For the object-centric Co3D dataset, we evaluate the accuracy of novel views using PSNR and LPIPS [53] metrics. Note that PSNR/LPIPS are not meaningful metrics for large scenes since the predictions have a large amount of uncertainty, i.e., a wide range of novel view images can be consistent with any input image. Thus, we report FID [54] and KID [55] scores to evaluate the realism of reconstructions in these cases. Our approach outperforms all baselines for LPIPS, FID, and KID metrics, as our model achieves more realistic results. We achieve slightly lower PSNR compared to pixelNeRF [9]. Note that PSNR favors mean estimates, and that we only evaluate our model using a single randomly sampled scene for an input image due to computational constraints.

### 4.2 Single-Image Motion Prediction

Here, we seek to train a model that, given a single static image, allows us to sample from *all possible motions* of pixels in the image. Given, for instance, an image of a person performing a task, such as kicking a soccer ball, it is possible to predict potential future states. This is a stochastic problem, as there are multiple possible motions consistent with an image. We train on a dataset of natural videos [56]. We only observe RGB frames and never directly observe the underlying motion, i.e, the pixel correspondences in time are unavailable. We use tuples of two frames from videos within a small temporal window, and use them as our context and target observations for training.

**3D Scene Completion**                                                **GAN Inversion**

| | | Co3D | | | RealEstate10k | | | | FFHQ | |
|---|---|---|---|---|---|---|---|---|---|---|
| | PSNR↑ | LPIPS↓ | FID↓ | KID↓ | FID↓ | KID↓ | | | FID↓ | KID↓ |
| pixelNeRF | **17.93** | 0.54 | 180.20 | 0.14 | 195.40 | 0.14 | Determ. | | 25.7 | 0.019 |
| SparseFusion | 12.06 | 0.63 | 252.13 | 0.16 | 99.44 | 0.04 | Ours | | **7.45** | **0.002** |
| Ours | 17.47 | **0.42** | **84.63** | **0.05** | **42.84** | **0.01** | | | | |

Table 1: **Quantitative evaluation.** (left) We benchmark our 3D generative model with state-of-the-art baselines pixelNeRF [9] and SparseFusion [5]. (right) We benchmark with a deterministic baseline on GAN inversion, which we drastically outperform.

**Related Work.** Several papers tackle this problem, where motion in the form of optical flow [57–59], 2D trajectories [60, 61], and human motion [62, 63] are recovered from a static image; however, all these methods assume supervision over the underlying motion. Learning to reason about motion requires the neural network to learn about the properties and behavior of the different objects in the world. Thus, this serves as a useful proxy task for representation learning, and can be used as a backbone for many downstream applications [60, 64].

**Structure of S and forward model `warp`.** Our signal $\mathbf{S}$ stores the appearance and motion information in a 2D grid. At any pixel $\mathbf{u}$, the signal is defined as $\mathbf{S}(\mathbf{u}) = (\mathbf{S}_c(\mathbf{u}), \mathbf{S}_m(\mathbf{u}))$, where $\mathbf{S}_c(\mathbf{u}) \in \mathbb{R}^3$ is the color value, and $\mathbf{S}_m(\mathbf{u}) \in \mathbb{R}^2$ is a 2D motion vector. The forward model is a warping operator, such that $\texttt{warp}(\mathbf{S}, \phi)(\mathbf{u} + \phi\mathbf{S}_m(\mathbf{u})) = \mathbf{S}_c(\mathbf{u})$ and $\phi$ is a scalar that changes the magnitude of motion. We implement this function using a differentiable point splatting operation [65].

**Implementation of `denoise`.** The inset figure illustrates our design. We use a 2D network that takes $\mathbf{O}^{\text{ctxt}}$, $\mathbf{O}_t^{\text{trgt}}$, and $t$ as input, and generates the motion map $\mathbf{S}_m$ as the output. The signal is then reconstructed as $\mathbf{S} = (\mathbf{O}^{\text{ctxt}}, \mathbf{S}_m)$. Context and target frames correspond to parameters $\phi^{\text{ctxt}} = 0$ and $\phi^{\text{trgt}} = 1$, and can be reconstructed from the signal using `warp`.

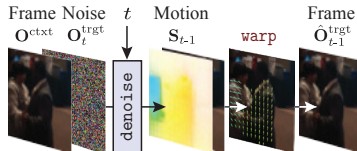

**Loss and Evaluation.** Similar to inverse graphics, we use reconstruction and regularization losses. The reconstruction losses are identical to Eqs. 8 and 9, and the regularization loss is a smoothness term that encourages a natural motion of the scene, see supplement for details. We show results in Fig. 5 (left), where we can estimate a diverse set of possible motion flows from monocular images. By smoothly interpolating $\phi$, we can generate short video sequences, even though our model only saw low-framerate video frames during training. We also train a deterministic baseline, which only generates a single motion field. Due to the amount of uncertainty in this problem, the deterministic estimate collapses to a near-zero motion field regardless of the input image, and thus, fails to learn any meaningful features from images.

### 4.3 GAN Inversion

Projecting images onto the latent space of generative adversarial networks is a well-studied problem [8, 66], and enables interesting applications, as manipulating latents along known directions allows a user to effectively edit images [67–69]. Here, we solve the problem of projecting partial images: given a small visible patch in an image, our goal is to model the distribution of possible StyleGAN2 [8] latents that agree with the input patch. There are a diverse set of latents that can correspond to the input observation, and we train our method without observing supervised (`image`, `latent`) pairs. Instead, we train on pairs of $(\mathbf{O}^{\text{ctxt}}, \mathbf{O}^{\text{trgt}})$ observations, where $\mathbf{O}^{\text{ctxt}}$ are the small patches in images, and $\mathbf{O}^{\text{trgt}}$ are the full images.

**Related Work.** While most GAN inversion methods focus on inverting a complete image into the generator's latent space [70–78], some also reconstruct GAN latents from small patches via supervised training. Inversion is not trivial, and papers often rely on regularization [77] or integrate the inversion with editing tasks [79] for higher quality. We also integrate the inpainting task with the inversion, and seek to model the uncertainty of the GAN inversion task given only a partial observation (patch) of the target image.

**Structure of S and forward model `synthesize`.** Our signal $\mathbf{S} \in \mathbb{R}^{512}$ is a 512 dimensional latent code representing the "w" space of StyleGAN2 [8] trained on the FFHQ [80] dataset. The forward model $\texttt{synthesize}(\mathbf{S}, \phi) = \texttt{GAN}(\mathbf{S})[\phi]$ first reconstructs the image corresponding to $\mathbf{S}$ using a

forward pass of the GAN. It then extracts a patch using the forward model's parameters $\phi$ that encode the patch coordinates.

**Implementation of `denoise`, Loss, and Evaluation.** Please see the inset figure for an illustration of the method. The denoising network receives $\mathbf{O}^{\text{ctxt}}$, $\mathbf{O}_t^{\text{trgt}}$, and timestep t as input, and generates an estimate of the StyleGAN latent $\mathbf{w}$. The loss function is identical to Eq. 8 and compares the reconstructed sample with ground truth.

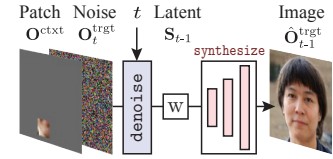

We show results in Fig. 5 (right). We obtain diverse samples that are all consistent with the input patch. We also compare with a deterministic baseline that minimizes the same loss but only produces a single estimate. While this deterministic estimate also agrees with the input image, it does not model the diversity of outputs. We consequently achieve significantly better FID [54] and KID [55] scores than the deterministic baseline, reported in Tab. 1 (right).

## 5 Discussion

**Limitations.** While our method makes significant advances in generative modeling, it still has several limitations. Sampling 3D scenes at test time can be very slow, due to the expensive nature of the denoising process and the cost of volume rendering. We need multi-view observations of training scenes for the inverse graphics application. Our models are not trained on very large-scale datasets, and can thus not generalize to out-of-distribution data.

**Conclusion** We have introduced a new method that tightly integrates differentiable forward models and conditional diffusion models. Our model learns to sample from the distribution of signals trained only using their observations. We demonstrate the efficacy of our approach on three challenging computer vision problems. In inverse graphics, our method, in combination with a 3D-structured conditioning method, enables us to directly sample from the distribution of real-world 3D scenes consistent with a single image observation. We can then render multi-view consistent novel views while obtaining diverse samples of 3D geometry and appearance in unobserved regions of the scene. We further tackle single-image conditional motion synthesis, where we learn to sample from the distribution of 2D motion conditioned on a single image, as well as GAN inversion, where we learn to sample images that exist in the latent space of a GAN that are consistent with a given patch. With this work, we make contributions that broaden the applicability of state-of-the-art generative modeling to a large range of scientifically relevant applications, and hope to inspire future research in this direction.

**Acknowledgements.** This work was supported by the National Science Foundation under Grant No. 2211259, by the Singapore DSTA under DST00OECI20300823 (New Representations for Vision), by the NSF award 1955864 (Occlusion and Directional Resolution in Computational Imaging), by the ONR MURI grant N00014-22-1-2740, and by the Amazon Science Hub. We are grateful for helpful conversations with members of the Scene Representation Group David Charatan, Cameron Smith, and Boyuan Chen. We thank Zhizhuo Zhou for thoughtful discussions about the SparseFusion baseline. This article solely reflects the opinions and conclusions of its authors and no other entity.

**Author contributions.** Ayush and Vincent conceived the idea of diffusion with forward models, designed experiments, generated most figures, and wrote most of the paper. Ayush contributed the key insight to integrate differentiable rendering with diffusion models by denoising in image space while generating 3D scenes. Ayush and Vincent generalized this to general forward models, and conceived the single-image motion application. Vincent contributed the 3D-structured conditioning and generated the overview and methods figures. Ayush wrote all initial code and ran all initial experiments. Ayush and Tianwei implemented the inverse graphics application and generated most of the 3D results of our model, while George helped with the baseline 3D results. Ayush executed all single-image motion experiments. George conceived, implemented, and executed all GAN inversion experiments. Semon helped formalizing the method and wrote the proposition and its proof. Frédo and Bill were involved in regular meetings and gave valuable feedback on results and experiments. Josh provided intriguing cognitive science perspectives and feedback on results and experiments, and provided a significant part of the compute. Vincent's Scene Representation Group provided a significant part of the compute, and the project profited from code infrastructure developed by and conversations with other members of the Scene Representation Group.

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
