# Diffusion with Forward Models: Solving Stochastic Inverse Problems Without Direct Supervision

**Ayush Tewari**[1][*]    **Tianwei Yin**[1][*]    **George Cazenavette**[1]    **Semon Rezchikov**[4]
**Joshua B. Tenenbaum**[1,2,3]    **Frédo Durand**[1]    **William T. Freeman**[1]    **Vincent Sitzmann**[1]

[1]MIT CSAIL    [2]MIT BCS    [3]MIT CBMM    [4]Princeton IAS

## Contents

## 1 Proposition

**Proposition 1.** Suppose that any signal $\mathbf{S}$ can be reconstructed from the set of all *all possible* observations of $\mathbf{S}$. Under this assumption, if in the limit as the number of known observations per signal goes to infinity, there are parameters $\theta$ such that $\mathcal{L}_{\theta}^{\mathrm{trgt}} + \mathcal{L}^{\mathrm{novel}}$ is minimized, then the conditional probability distribution over signals discovered by our model $p(\mathbf{S} \mid \mathbf{O}^{\mathrm{ctxt}}; \phi^{\mathrm{ctxt}})$ agrees with the true distribution $p^{\mathrm{true}}(\mathbf{S} \mid \mathbf{O}^{\mathrm{ctxt}}; \phi^{\mathrm{ctxt}})$.

The total observation loss is defined in Equation equation 4 below.

After introducing some notation, we will formalize the assumptions made in the proposition.

**Definition 1.** We call the collection of all observations that correspond to a signal a *total observation* of the signal $\mathbf{O}^{\mathrm{total}}$. Formally,

$$\mathbf{O}^{\mathrm{total}} = \mathbf{O}^{\mathrm{total}}(\mathbf{S}) = \{(\phi, \mathtt{forward}(\mathbf{S}, \phi))\}_{\phi \in \mathcal{P}}.$$

Here, $\mathcal{P}$ denotes the set of *parameters* of the forward model, e.g. $\mathcal{P} = SE(3)$ for the inverse graphics application in the paper.

**Definition 2.** We define the *scattering map* as the (measurable) map sending signal $\mathbf{S}$ to its total image $\mathbf{O}^{\mathrm{total}}$:

$$Scatter : \mathbf{S} \mapsto \mathbf{O}^{\mathrm{total}}(\mathbf{S}).$$

For a reference for the technical notion of a measurable map, see any textbook on measure theory (e.g. [1]); all maps arising in machine-learning models are measurable because they are piecewise continuous.

---

[*] Equal Contribution.

37th Conference on Neural Information Processing Systems (NeurIPS 2023).

**Assumption**   We formalize the assumption of Proposition 1 by requiring that there is a (measurable) map $Scatter^{-1}$ from total observations to signals which satisfies, for all signals under consideration,

$$Scatter^{-1}(Scatter(\mathbf{S})) = \mathbf{S}. \tag{1}$$

In other words, given all possible observations of a signal, we can uniquely reconstruct the signal (for the class of signals under consideration). Alternatively, the map $Scatter$ is injective. This assumption is a basic assumption necessary for many algorithms in 3D computer vision, and underlies the recent success of differentiable rendering for 3D scene reconstruction [2] from large sets of image observations. Note that there may be total observations $\mathbf{O}^{\text{total}} = \{(\phi, \mathbf{O}_\phi^{\text{total}})\}_{\phi \in \mathcal{P}}$ which do *not* arise as the total observations $\mathbf{O}^{\text{total}}(S)$ of any signal $\mathbf{S}$. Equation 1 makes no assumption on the behavior of $Scatter^{-1}(\mathbf{O}^{\text{total}})$ on such 'inconsistent' total observations $\mathbf{O}^{\text{total}}$.

**Observations generated by our model are slices of total observations.**   A basic property of our model is that the target observations arise from predicted signals, since

$$\mathbf{O}^{\text{trgt}} = \texttt{forward}(\mathbf{S}, \phi^{\text{trgt}}).$$

Thus, our model is limited to modeling the space over observations that are a member of the total observations set, i.e., $(\phi^{\text{trgt}}, \mathbf{O}^{\text{trgt}}) \in \mathbf{O}^{\text{total}}(\mathbf{S})$ for some signal $\mathbf{S}$. This is an important property that is not trivially true for many existing models, e.g., for inverse graphics, many light-field-based approaches [3–6] do not satisfy this property.

**The predicted distribution over signals can be recovered from the distribution over observations.** Since we can reconstruct a signal from its total observation, we have that $Scatter^{-1}(Scatter(U)) = U$ for any set of signals $U$. Writing

$$V = \{(\phi^{\text{trgt}}, \texttt{forward}(\mathbf{S}, \phi^{\text{trgt}})) \mid \phi^{\text{trgt}} \in \mathcal{P}, \mathbf{S} \in U\} = \{Scatter(\mathbf{S}) \mid \mathbf{S} \in U\}.$$

for the set of *total observations* of signals $\mathbf{S} \in U$, we therefore have that

$$p(\mathbf{O}^{\text{total}}(\mathbf{S}) \in V \mid \mathbf{O}^{\text{ctxt}}; \phi^{\text{ctxt}}) = p(\mathbf{S} \in U \mid \mathbf{O}^{\text{ctxt}}; \phi^{\text{ctxt}}). \tag{2}$$

As such, we can recover $p(\mathbf{S} \in U \mid \mathbf{O}^{\text{ctxt}}; \phi^{\text{ctxt}})$ by computing $p(\mathbf{O}^{\text{total}}(\mathbf{S}) \in V \mid \mathbf{O}^{\text{ctxt}}; \phi^{\text{ctxt}})$ for all possible $V$ (where we note that if $V$ consists of total observations that do not arise from signals then its probability is zero).

**Our loss maximizes the likelihood over total observations.**   We now claim that the loss we optimize forces our model to find parameters $\theta$ such that

$$p(\mathbf{O}^{\text{total}}(\mathbf{S}) \in V \mid \mathbf{O}^{\text{ctxt}}; \phi^{\text{ctxt}}) = p^{true}(\mathbf{O}^{\text{total}}(\mathbf{S}) \in V \mid \mathbf{O}^{\text{ctxt}}; \phi^{\text{ctxt}}) \tag{3}$$

We first define the *total observation loss*

$$\mathcal{L}_\theta^{\text{total}} = \mathbb{E}_{\mathbf{O}^{\text{ctxt}}, \mathbf{O}^{\text{trgt}}, \phi^{\text{ctxt}}, \phi^{\text{trgt}}, t} \left[ \|\mathbf{O}^{\text{trgt}} - \texttt{forward}(\texttt{denoise}_\theta(\mathbf{O}^{\text{ctxt}}, \mathbf{O}_t^{\text{total}}; t, \phi^{\text{ctxt}}), \phi^{\text{trgt}})\|^2 \right] \tag{4}$$

This is the same as $\mathcal{L}_\theta^{\text{target}}$ of the main text, but with $\texttt{denoise}_\theta$ depending on the total observation. We now have the identity

$$\mathbb{E}_{\mathbf{O}^{\text{trgt}}, \phi^{\text{trgt}}} \left[ \|\mathbf{O}^{\text{trgt}} - \texttt{forward}(\texttt{denoise}_\theta(\mathbf{O}^{\text{ctxt}}, \mathbf{O}_t^{\text{total}}; t, \phi^{\text{ctxt}}), \phi^{\text{trgt}})\|^2 \right]$$
$$= \|\mathbf{O}_t^{\text{total}} - \hat{\mathbf{O}}_{t-1}^{\text{total}}\|^2 \tag{5}$$
$$= C_t D_{KL}(q(\mathbf{O}_{t-1}^{\text{total}} \mid \mathbf{O}_t^{\text{total}}, \mathbf{O}_0^{\text{total}}, \mathbf{O}^{\text{ctxt}}; \phi^{\text{trgt}}) \mid p_\theta(\mathbf{O}_{t-1}^{\text{total}} \mid \mathbf{O}_t^{\text{total}}, \mathbf{O}^{\text{ctxt}}; \phi^{\text{ctxt}}).$$

where $C_t$ is some positive constant for each $t$; this follows from Equations 95-99 of [7]. Thus, *if the model has parameters $\theta$ such that Eq. 3 holds for this parameter, then this will also hold the global minimum of the loss,* since Eq. 3 holds exactly when the $t = 1$ term of Eq. 5 is zero.

It is natural to train such a model by minimizing the loss

$$\|\mathbf{O}^{\text{trgt}} - \underbrace{\texttt{forward}(\texttt{denoise}_\theta(\mathbf{O}^{\text{ctxt}}, \mathbf{O}_t^{\text{total}}; t, \phi^{\text{ctxt}}), \phi^{\text{trgt}})}_{=\hat{\mathbf{O}}_{t-1}^{\text{trgt}}}\|^2$$

with randomly-sampled forward-model parameters $\phi^{\text{trgt}}$. This is what we do in our real training procedure, except that $\texttt{denoise}_\theta$ now only depends on a slice of $\mathbf{O}_t^{\text{total}}$, namely $\mathbf{O}_t^{\text{trgt}}$, as well as on $\phi^{\text{trgt}}$, see Eq. 8 in the main paper. Now, nothing in equation 5 requires $\hat{\mathbf{O}}_{t-1}^{\text{total}}$ to depend on all of $\mathbf{O}_t^{\text{total}}$; the equation is still valid even if $\hat{\mathbf{O}}_{t-1}^{\text{total}}$ is a function only of a slice of $\mathbf{O}_t^{\text{total}}$. This is precisely the case in our training procedure. As such, the addition of the term $\mathcal{L}_{\text{novel}}$ to the loss (see Eq. 9 in the main paper), when $\phi^{\text{novel}}$ is stochastically sampled, *forces the quantity in Equation 5 to be minimized* even though our estimate of the denoised signal only depends on the context observation and the noised target observation. Thus, the conclusion of this proposition still applies to our training procedure.

**Conclusion of proof.** We now conclude that

$$p(\mathbf{S} \in U \mid \mathbf{O}^{\text{ctxt}}; \phi^{\text{ctxt}}) = p^{true}(\mathbf{S} \in U \mid \mathbf{O}^{\text{ctxt}}; \phi^{\text{ctxt}}) \tag{6}$$

by applying (3) and using the fact that (2) holds both for $p$ and for $p^{true}$. Equation 6 is the desired conclusion.

**Remark on 3D consistency.** Technically, in the models in our paper, we have that $p(\mathbf{S} \mid \mathbf{O}^{\text{ctxt}}; \phi^{\text{ctxt}})$ is actually dependent on auxiliary parameter $\phi^{\text{trgt}}$: we make predictions over signals using a diffusion model coupled with a *particular choice* of forward-model parameter. As such, to be precise, we say that our model predicts a family of distributions $p_{\phi^{\text{trgt}}}(\mathbf{S} \mid \mathbf{O}^{\text{ctxt}}; \phi^{\text{ctxt}})$ depending on $\phi^{\text{trgt}}$, where these distributions may differ for different values of $\phi^{\text{trgt}}$. Correspondingly, the model predicts a family of distributions $p_{\phi^{\text{trgt}}}(\mathbf{O}^{\text{total}} \mid \mathbf{O}^{\text{ctxt}}; \phi^{\text{trgt}})$. However, the addition of the $\mathcal{L}_{\text{novel}}$ term forces learned distribution over signals to agree with the true distribution over signals in the limit of infinite observations; as such, in that same limit, the learned distribution over signals becomes independent of $\phi^{\text{trgt}}$ since the true distribution is manifestly independent of it.

**Inverting the scatter map is unnecessary.** In the above argument, while we assumed the inverse to the $Scatter$ map, we did not need to *compute* the map $Scatter^{-1}$ to argue that the estimated probability densities $p(\mathbf{S} \mid \mathbf{O}^{\text{ctxt}}; \phi^{\text{ctxt}})$ agree with $p^{true}(\mathbf{S} \mid \mathbf{O}^{\text{ctxt}}; \phi^{\text{ctxt}})$. This is a highly desirable property, as the map $Scatter^{-1}$ often cannot be computed efficiently. Thus, our model learns correct estimates of $p^{true}(\mathbf{S} \mid \mathbf{O}^{\text{ctxt}}; \phi^{\text{ctxt}})$ without ever explicitly computing $Scatter^{-1}$.

## 2 Details of the Method

### 2.1 Inverse Graphics

**Loss Function** We incorporate the use of several regularization terms:

$$\mathcal{L}_{\text{reg}} = \mathcal{L}_{\text{LPIPS}} + \mathcal{L}_{\text{depth}} + \mathcal{L}_{\text{cond}}, \tag{7}$$

$$\mathcal{L}_{\text{LPIPS}} = \mathcal{L}_{\text{LPIPS}}(\hat{\mathbf{O}}_{t\text{-}1}^{\text{trgt}}, \mathbf{O}^{\text{trgt}}), \tag{8}$$

$$\mathcal{L}_{\text{depth}} = \mathcal{L}_{\text{EAS}} + \mathcal{L}_{\text{dist}}, \tag{9}$$

$$\mathcal{L}_{\text{cond}} = \|\mathbf{O}_{\text{det}}^{\text{trgtcolor}} - \mathbf{O}^{\text{trgt}}\|^2. \tag{10}$$

Here, $\mathcal{L}_{\text{LPIPS}}$ is the LPIPS perceptual loss [8] that encourages rendered images to be perceptually similar to the ground truth observation. This has been shown to help improve the quality of diffusion models [9]. We further regularize the depth renderings from the target and novel viewpoints using an edge-aware smoothness loss [10] and a distortion loss [11] that discourages floating geometry artifacts. These depth regularization terms encourage *natural* 3D geometry reconstructions. Finally, we use $\mathcal{L}_{\text{cond}}$ on the rgb component of the deterministic estimate $\mathbf{O}_{\text{det}}^{\text{trgt}}$, denoted as $\mathbf{O}_{\text{det}}^{\text{trgtcolor}}$. Recall that we use $\mathbf{O}_{\text{det}}^{\text{trgt}}$ to condition our denoising network, and that it includes color as well as high-dimensional features. We use multiplier hyperparameters $0.2$ for $\mathcal{L}_{\text{LPIPS}}$ and $0.02$ for $\mathcal{L}_{\text{depth}}$. Our code will be publicly released to aid in reproducibility.

**Remark on regularization** Recall our assumption in Proposition 1 that the map from all observations to the signal is invertible. In the 3D setting, where we use real-world 2D datasets for training, we do not have access to all possible signal observations. On such training data, this assumption is not strictly true, since multiple 3D scenes can explain a subset of observations. However, the addition of the regularizing terms singles out *preferred* choices of 3D scenes explaining the known observations.

Heuristically, our depth and smoothness regularizers $\mathcal{L}_{\mathrm{depth}}$ make the map between scenes and the observations in the training dataset invertible, i.e., they single out a *uniquely determined natural* 3D scene that explains the observations in the impoverished 2D dataset.

**Training Details** Volume rendering is an expensive computation, making training our 3D models under limited memory budgets challenging. However, unlike image-space diffusion models, where the entire image is predicted directly, we can render pixels independent of each other using `render`. In practice, we render $24 \times 24$ patches at random positions in the image in each training iteration. We use the vision transformer architecture from DiT [12] to implement the image backbone `enc` in `denoise`. We modify the MLP architecture (`MLP`) of pixelNeRF to support additional time conditioning input in our models. Our models are trained on 8 A100 GPUs, with a batch size of 24. Training takes around 7 days for RealEstate10k, and around 3 days for Co3D. We use ADAM with a learning rate of $2e-5$. We initially train at a resolution of $64 \times 64$. We then finetune the model at $128 \times 128$. For our Co3D models, we found it helpful to first pretrain the deterministic conditioning component of the model for 10k iterations. We use 64 samples each for coarse and fine stages for volume rendering of the output 3D reconstruction, and only 32 coarse samples for rendering the conditioning input.

We process the Co3D dataset following [5], i.e., we center-crop the images and resize them to a consistent resolution. We follow Chan et al. [5] to provide the absolute pose of the input image as an additional input to the encoder. We also use this input for our baselines, except when using official codebases of SparseFusion [13] and pixelNeRF [14]. During training, we randomly select the initial context frame, and pick a target frame for denoising within predetermined distance intervals. We randomly choose one additional frame between initial context frame and the target frame for computing the novel view reconstruction loss ($\mathcal{L}_{\mathrm{novel}}$). To support autoregressive sampling, we add more context frames from the dataset during training, such that the network can reason jointly from multiple input images. At test time, we iteratively sample new images that are then added as a context frame for the next frame. Autoregressive sampling allows us to cover the entire 360 regions in Co3D scenes by diffusing multiple images around the object. We follow the same training strategy for RealEstate10k, except that we do not feed in absolute poses to our encoder or the baselines. We augment the RealEstate10k dataset by randomly reversing the order of frames in the videos.

**Baselines** We use code provided by the authors for SparseFusion and train on our datasets. We note that the SparseFusion paper only demonstrated results on segmented-out objects without any background, and used multiple input images at test time, unlike our monocular method. Since SparseFusion uses a pretrained VAE backbone that only takes inputs at $256 \times 256$ resolution, we train it at this resolution. However, the 3D optimization is performed at the same resolution as our method. We use the official repository (https://github.com/sxyu/pixel-nerf) for the pixelNeRF baseline. We use 50 scenes for Co3D and 100 scenes for RealEstate for the quantitative evaluations.

## 2.2 Single-Image Motion Prediction

We use an edge-aware smoothness regularization loss on the motion field that is equivalent to the smoothness loss on the depths defined in Sec. 2.1. We use the DiT archicture as our denoising model. The clean context image and the noisy target image are concatenated along the channel dimension and used as input to the network. The output is pixel-aligned motion field that is used to warp the context into the target using SoftSplat [15]. We use ADAM with a learning rate of $2e - 5$ and batch size of 72 to optimize our networks on 2 RTXA6000 GPUs. Our models are trained on the Vimeo90K dataset [16].

## 2.3 GAN Inversion

The GAN into which we are inverting is a StyleGAN2-Ada [17] trained on the $256 \times 256$ FFHQ dataset [18]. Our "ground truth" target dataset is generated by taking random samples from this GAN with a truncation $\psi$ of 0.5, down-sampled to $64 \times 64$ pixels. We, again, use the DiT architecture as our denoising model. The context images are obtained by taking a ground truth sample and masking out all but a small patch of varying size. The masked context image and noise target image are concatenated along the channel dimension and used as input to the denoising network, which predicts the denoised "$\mathbf{w}$" code. This $\mathbf{w}$ is then fed through the forward model (generator) and downsampled to

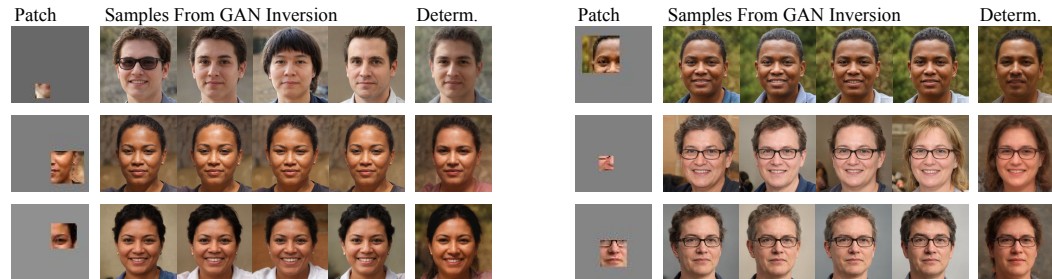

Figure 1: More samples from our GAN Inversion model. Our method produces many plausible faces given only a small patch.

$64\times64$ pixels to obtain our denoised target image. All of our training is done at $64\times64$ resolution, but using a GAN trained on high-resolution $256\times256$ images allows us to obtain high-resolution results at test time by simply not downsampling the final denoised output. We use the ADAM optimizer with a learning rate of 2e-5 and a batch size of 4 to train our networks on a single RTXA6000 GPU. We include more results in Figure 1.

## 2.4 Sampling

We use 50 DDIM [19] denoising timesteps for all our results across all applications. All our models, except the inverse graphics model trained on RealEstate10k, are trained without any classifier-free guidance. For our RealEstate10k model, we use a classifier-free guidance weight of 2. Here, the model is also trained as an unconditional model, where the conditioning image is zeroed out for 10% of the iterations.

## 3 Limitations

While we present the first method that enables diffusion models to learn the conditional distribution over signals, only using observations through a forward model, our approach has several limitations. Our sampling times can be very expensive in some cases. The sampling time ranges from just a few seconds for our GAN application, to around 100 mins for 360-degree autoregressive sampling for Co3D. This is both due to the expensive nature of the iterative denoising process, as well as the cost of rendering 3D reconstructions using volume rendering. Our training has large memory requirements, and can thus not be trained on smaller GPUs. Future work on making these models easier to train would make them mode applicable. Our models are not trained on very large-scale datasets, and can thus not generalize to out-of-distribution data. Finally, we only present preliminary investigations into applications outside inverse graphics; however, we hope that we offer a strong experimental base that can be beneficial for future exploration.