# OpenReview forum: "Diffusion with Forward Models: Solving Stochastic Inverse Problems Without Direct Supervision"
_NeurIPS.cc/2023/Conference — NeurIPS 2023 spotlight_

### Official Review · Reviewer_zgt6 · 2023-07-04

**Soundness:** 2 fair
**Presentation:** 1 poor
**Contribution:** 3 good
**Rating:** 5
**Confidence:** 3

**Summary:**

The authors present a new method for training diffusion models based on partial observations of underlying signals. Based on a dataset containing groups of multiple partial observations of the same signal, as well as a differentiable model of the forward operator, this method can train a diffusion model for synthesizing underlying signals conditioned on a given partial observation. The underlying signals can then be projected into "novel views". They apply their method to inverse graphics, GAN inversion, and motion prediction. The method outperforms the tested baselines in several metrics.

**Strengths:**

- Results seem strong, both visually and quantitatively.
- Paper is well-motivated, and addresses an important problem.
- The metrics used to evaluate performance are sound and relevant.

**Weaknesses:**

 - Proposition 1 is very vaguely written. It needs to be presented with more rigorous definitions (what are the assumptions in math form? what is the result? "agrees with" is not formal enough and the "if" statement is not clear to me).
- More generally, the method section is presented in simplified terms. There needs to be more math there. Do you prove that your proposed loss function is indeed related to the maximum likelihood objective? KL divergence? What are the conditions (in math form) for Proposition 1? How does the sampling algorithm (after training) work? The overall presentation of the experiments section is also overly convoluted and unclear.
- The requirement on the dataset to have multiple views of the same scene is still a significant assumption over the dataset. While this might make sense in 2D-to-3D, it makes less sense in other stochastic inverse problems such as accelerated MRI reconstruction (we almost never have two scans of the same object/person).

Minor issues:
- In lines 166-170, diffusion-based inverse problem solvers are mentioned. The main weakness of those works that is mentioned is their inability to learn from partial observations. It would be beneficial to discuss concurrent works [W1, W2] that model signals based on partial observations as well. What are the similarities and differences with your work?

[W1] https://arxiv.org/abs/2305.13128

[W2] https://arxiv.org/abs/2305.19256

**Questions:**

- How does your work compare to the many NeRF+Diffusion papers? Examples include [Q1, Q2] but there are many more. I ampositive that some of them apply to the same use case the authors present, and can be added as entries in Table 1.

[Q1] https://arxiv.org/abs/2304.06714

[Q2] https://arxiv.org/abs/2304.14473

**Limitations:**

No, the authors have not adequately addressed the limitations and potential negative societal impact of their work.

---

> ### Author Rebuttal · Authors · 2023-08-10
>
> We thank the reviewer for the detailed comments. The reviewer notes that our “results seem strong, both visually and quantitatively”, we “outperform the tested baselines”, and that our paper “addresses an important problem”. We now address the remaining concerns:
>
> ### Proposition
> We **intentionally** only provided a simplified version of the proposition in the main paper to preserve readability. **Note that we provide a rigorous statement and in-depth proof in the supplemental material** (Paper line 147: “...as we formally prove in the supplement:”). Following your comments, we will make the statement in the paper more rigorous and provide an abridged version of the proof already in the main paper. We will also make the references to the supplemental sections more prominent.
>
> ### Loss Function
> In the supplemental material, we **formally relate our loss function to the maximum likelihood objective**; see our supplemental Sec. 2 (“Our loss maximizes the likelihood over total observations”). Following your comment, we will make this explicitly clear in the main paper and reference the supplemental when discussing the loss.
>
> ### Sampling at test time
> Given a context observation, we start from pure noise and denoise the underlying signal and target observation, see L138-139 (“At test time, a signal is sampled by iterating Eq. 4, 5, and 6, ….”).  Also, see Figure 1 (bottom left “Single Denoising Step”). We iterate the single denoising step depicted here, starting from O^{trgt}_T. We will improve the exposition of the sampling stage and make it more prominent. For the inverse graphics application, refer to the updated overview figure; see general comment (G5).
>
> ### Presentation of the experiments
> We provide details for each application in the supplemental document; see Section 3. We will work on making the exposition clearer and release our code to aid in understanding and reproduction. We have already updated the overview figure for the inverse graphics application; see general comment (G5).
>
> ### Multiple views required for training
> We require multiple observations during training; however, they do not necessarily need to cover the signal completely. Consider the 3D reconstruction application: During training, we only observe sparse camera views, observing only a small subset of each scene. Yet, our model can reconstruct complete scenes at test time, such as a complete 360 view of an object, by denoising the test camera trajectory.
>
> To further demonstrate this, we conducted an experiment where we learn to reconstruct 64x64 res. images conditioned on partial observations that have a 32x32 patch missing. Referencing our formulation, the signal is the complete 2D image, and the forward model is a sampling function that can sample a specific region from the complete image.
> Importantly, we only use **incomplete** images during training, i.e., one patch from the image is always missing in our training dataset. From such training data, we first compute our context and target observations by applying further degradation to our training images - we remove pixels from the training image to compute the context, and the removed pixels become the target.
>
> Our method now learns to sample complete signals conditioned on the context observations. We use a similar architecture as our 3D reconstruction pipeline (see updated figure (G5)), where we condition the diffusion model on a deterministic estimate computed from the context. This network takes the context as input and computes a deterministic estimate of the signal we supervise only at the visible regions in the training images. The diffusion model is conditioned on this deterministic estimate and learns to sample the target patch by first reconstructing the signal, i.e., the complete image, and applying the forward model. At test time, we can complete the missing information in any image; see the response PDF. This shows that our method could likely apply to a wide range of stochastic inverse problems, including medical imaging. We also compare to the determinstic baseline in the response PDF.
>
> ### Relation to concurrent works that learn from partial data
> As correctly noted, they are concurrent to our submission and appear on arxiv after the submission deadline. The main difference is that both these papers seem to assume a linear forward model, while our approach supports a much wider range of forward models, as we demonstrate in our experiments. Our differentiable renderer is non-linear and GAN inversion also involves a non-linear forward pass of the GAN model.
>
> ### Relation to NeRF+diffusion papers
> We discuss many papers, including concurrent works, that use NeRF and diffusion, in Sec. 3 and Sec. 4.1.  The most closely related paper, in terms of the nature of the model, is RenderDiffusion, see general discussion comment (G1). The most closely related paper in terms of the task and the complexity of datasets is SparseFusio, which we extensively compare to and significantly outperform.
>
> We briefly discuss the two papers:
> Both [Q1] and [Q2] train 3D diffusion models using purely 3D architectures, adding noise to some 3D latents that represent the scene during training. As these 3D latents are not available when training from 2D observations, they either have to be pre-computed [Q2] or discovered jointly [Q1]. Our approach is radically different, where we show that learning to denoise observations can lead to sampling in 3D. We do not add noise to any unknown quantity at training time. Instead, we add noise to the known observation space.
>
> Further, we show that our model scales to significantly more complex scenes, such as the compositional real indoor scenes from RealEstate10k, compared to the limited complexity of scenes considered in these two papers.
>
> Please also note that both these papers are concurrent to our submission.
>
> Thanks for pointing out all the references. We will cite them.
>
> ### Limitations
> See general discussion point (G4).

---

> > ### Comment · Reviewer_zgt6 · 2023-08-12
> >
> > I appreciate the authors' efforts in the rebuttal. I trust the authors to include more rigorous definitions of their propositions (not necessarily proofs, just a complete set of assumptions and results) in the main paper, as well as a limitations section in the main paper.
> > The limitations section should also mention the requirement on multiple views of the same scene. I understand this does not cover the scene completely, but it is nevertheless different than having one view per scene.
> >
> > Based on the rebuttal, I raise my score to 5.

---

> > > ### Author Response · Authors · 2023-08-15
> > >
> > > Thank you very much for considering our rebuttal! We are happy that we could answer your core questions. We will make the suggested changes in our paper.
> > >
> > > Borderline accept seems to imply that there might be ways to make the paper stronger. Could you kindly mention in what ways we could make the paper stronger yet?

---

### Official Review · Reviewer_LcVC · 2023-07-06

**Soundness:** 4 excellent
**Presentation:** 4 excellent
**Contribution:** 4 excellent
**Rating:** 8
**Confidence:** 4

**Summary:**

This work proposes a framework that allows learning a conditional diffusion model for a signal without direct observations. The proposed method integrates differentiable forward models with conditional diffusion models. It is evaluated for the task of learning image-conditioned 3D scenes from 2D images with poses and a few more downstream tasks.

**Strengths:**

1. The proposed method addresses an important problem. A main limitation of diffusion models is that they typically require direct observations on the signal of interest. The proposed method showed promising results on learning with indirect observations only.
2. The proposed method is novel (by considering RenderDiffusion as concurrent), which extends the classical diffusion process. The network architecture is well-designed and intuitive.
3. The effectiveness of the proposed method is demonstrated on multiple tasks and datasets, making the results comprehensive and solid.

**Weaknesses:**

I did not find any major weakness in this paper.
A minor weakness: on CO3D dataset, only hydrants category is evaluated. It could make the results stronger to show the performance on more categories.
Some other points are put into Questions section, which may make the submission stronger if they are clarified.

**Questions:**

1. If the context is empty, i.e. making the model unconditional, is proposed method mathematically equivalent to [1] RenderDiffusion? If not, what are the major differences in formulations?
2. In line 137 equation (4) (5) (6), to my understanding, \hat{O} is the "predicted x0" in DDPM, it is an estimate of clean observation but in theory not an instance in the space of clean observations. \hat{O} is rendered from S_{t-1}, while a true clean observation has a corresponding neural field, is it a practical fact or something provable that an "estimated observation" \hat{O} has a corresponding neural field S_{t-1}?

[1] RenderDiffusion: Image Diffusion for 3D Reconstruction, Inpainting and Generation

**Limitations:**

Yes

---

> ### Author Rebuttal · Authors · 2023-08-10
>
> We thank the reviewer for their careful read and insightful comments. The reviewer notes that our method “addresses an important problem” and shows “promising results”, our architecture is “well-designed and intuitive”, and our results are “comprehensive and solid’.  We now address remaining concerns:
>
> ### Evaluation on more Co3D categories
>
> Thank you for the suggestion. We evaluate the suggested setting, training a single model on all 10 Co3D categories. See general discussion point (G3) for details, and the rebuttal PDF for results. This evaluation goes beyond the category-specific training used in other existing papers, such as SparseFusion, which require training a single model per class. While our model generates plausible results, the diffusion model in SparseFusion often fails to generate reasonable images, even generating output images from a different object category. Score distillation with such a model often fails to reconstruct any reasonable object. We further conducted an experiment where train on a subset of the large-scale object-centric Objaverse dataset. Both these experiments show that our models are scalable and effective.
>
> ### Differences to RenderDiffusion
> Even if the context is empty, our model is different from RenderDiffusion, as it does not rely on global pose information, i.e. assuming that all objects are aligned with one canonical coordinate frame, and does not rely on monocular supervision. We rely on the multi-view losses for our proof in the supplemental. See general comment (G1) for a more detailed discussion.
>
> ### Does every “estimated observation” (\hat{O})  have a corresponding signal (S_{t-1})?
> If we understand this right, the reviewer raises the point that while the “true” clean observation has a corresponding signal, is it necessarily true that the estimate of this clean observation \hat{O} also has a corresponding signal? In our models, \hat{O} is generated using a forward model operation on the estimate of the signal, and thus, the diffusion model is strictly limited to estimating \hat{O} that have a corresponding signal. If the estimated observation does not have any corresponding signal, we would not be able to properly minimize our loss functions. Thus, it is a practical fact that, for the forward models we describe in the paper, the estimate of the observation can also be described as a signal, letting us successfully optimize our networks. Thanks for the insightful question!

---

> > ### Comment · Reviewer_LcVC · 2023-08-19
> >
> > Thanks for the response! My questions are well addressed and I tend to keep my rating.

---

### Official Review · Reviewer_bLEB · 2023-07-06

**Soundness:** 3 good
**Presentation:** 3 good
**Contribution:** 3 good
**Rating:** 6
**Confidence:** 4

**Summary:**

This paper incorporates a differentiable forward model (e.g., rendering function) into a denoising probabilistic process in order to sample from distributions of underlying signals consistent with partial observations. For example, in inverse graphics the proposed approach would enable sampling from the distribution of 3D scenes consistent with a single 2D image.

The efficacy of the approach is showcased in three applications: sampling from the distribution of 3D scenes using only 2D images at train and test times; single image motion prediction (where the forward model is a warping operation); and projecting partial images onto the latent space of a GAN (GAN inversion).


**Strengths:**

**S1.** Diffusion models for sampling 3D shapes or scenes have only been explored recently. A key challenge in this direction is the lack of 3D training data. The method proposed in this paper is able to train 3D models without the need for 3D training data.

**S2.** Tackling Stochastic Inverse Problems more generally is a great way to broaden interest in advances in diffusion models.

**Weaknesses:**

**W1.** There are some (very) recent works that tackle the same problem tackled in this paper, e.g., (Karnewar et al. CVPR 2023), (Kim et al. CVPR 2023). It would be important for the authors to discuss their work in relation to such recent works.

**W2.** The evaluation is limited: few tasks, few baselines (e.g., compare with the more detailed evaluation in baseline SparseFusion [50]). In fact, for single-image motion prediction and GAN inversion the main result is in the form of predictions for two inputs.

**W3.** While the approach was framed as a solution to Stochastic Inverse Problems in general, all applications considered are in computer vision. In my view, the generality of the problem statement is a strength of the submission and a demonstration of this generality would have greatly strengthened the submission.

### References

Karnewar et al. HOLODIFFUSION: Training a 3D Diffusion Model using 2D Images, CVPR 2023.

Kim et al. NeuralField-LDM: Scene Generation with Hierarchical Latent Diffusion Models. CVPR 2023.


**Questions:**

**Q1.** How does your approach relate / differ w.r.t. the recent methods in W1 above?

**Q2.** What are the main limitations of the approach, e.g., complexity of training and inference?

**Limitations:**

No limitations or negative societal impact were discussed in the main submission.

---

> ### Author Rebuttal · Authors · 2023-08-10
>
> We thank the reviewer for their comments and questions. The reviewer notes that our paper tackles a “key challenge” in training 3D diffusion models, and that our general formulation is a “great way to broaden interest in advances in diffusion models”. We now address the concerns:
>
> ### HoloDiffusion [Karnewar et al.] and NeuralField-LDM [Kim et al.]:
> We address this remark in the following points:
> - Please note that we already discuss these papers in the original submission (HoloDiffusion in L162-163, NeuralField-LDM in L190) and talk about how they relate to our work. **However, we hear the reviewer in that this discussion may not cover all essential points - we will thus significantly expand on this discussion in the camera-ready submission as below.**.
> - Note that both of these papers are concurrent to our submission and do not constitute prior work.
> - While certainly related and worth discussing, their contributions do not significantly overlap with our contributions and they do not take away from the significance of our results. We discuss this in-depth in the following, **and will extend the discussion of these papers in the revised version of our paper**.
>     - HoloDiffusion is an unconditional 3D generative model limited to simple object-centric 3D scenes. Specifically, it can only be trained on pre-segmented (background-free) Co3D objects, one class at a time. HoloDiffusion adds noise to deterministic estimates of 3D scenes. This input differs at training and at test time, inducing a domain shift, which the authors address by introducing a 2-stage approach. In contrast, our model does not suffer from a domain shift at training and at test time, meaning that we can train our model end-to-end and in a single stage, without any multi-stage pipeline. Our model can further be trained on the much more complex class of indoor rooms from the RealEstate10k dataset, and further can model Co3D objects without pre-processing via segmentation and background removal. We present theoretical insights in our paper that clearly show how our formulation lets us optimize for 3D scenes using our 2D loss functions. As shown in our paper, we can learn distributions over complex compositional scenes that go far beyond the results of HoloDiffusion.
>     - NeuralField-LDM is a 2-stage approach, where 3D latents are first computed from the 2D data, and then a 3D diffusion model is trained on the 3D latents. Our approach is radically different, demonstrating that we can directly learn to sample in 3D by learning from 2D data without requiring computation of an intermediate stage of 3D latents.
>
>
> ### Limited Evaluation
> We believe that evaluations presented in our paper are detailed and demonstrate the effectiveness of the approach. This is acknowledged by 5usH (“experiments… demonstrate the effectiveness of the proposed method”), LcVC (“results (are) comprehensive and solid”), and zgt6 (“Results seem strong, both visually and quantitatively”). We now expand on the points raised by the reviewer.
>
> We evaluate against the state of the art in both deterministic and probabilistic reconstruction, pixelNeRF and SparseFusion, respectively. The reviewer points out the evaluation in SparseFusion, which features more deterministic baselines. Please note that pixelNeRF outperforms all other deterministic baselines in the SparseFusion evaluations (see their Table 2 PSNR). Thus, we believe that comparing with SparseFusion and pixelNeRF sufficiently demonstrates the quality of our method.
>
> While SparseFusion only uses the Co3D dataset for evaluation, we also use the very challenging scene-level RealEstate10k dataset. Nevertheless, we provide even more evaluations of our 3D reconstruction in this rebuttal where we train a general model on 10 categories of the Co3D dataset, see general discussion point (G3) and the rebuttal PDF.
>
>
> For the single-image motion and GAN inversion experiments, we do provide more results than the two examples that the reviewer mentions:
> We provide additional results in the supplemental pdf and the webpage.
> For the GAN inversion experiment, we provide quantitative results in Table 1 demonstrating improvements over a deterministic baseline. We further mention that the deterministic baseline for the single-image motion problem collapses (L290 - 291), thus making any quantitative evaluation meaningless.
> We are not aware of any other probabilistic baselines that we could compare to. For example, all encoder-based GAN inversion approaches we are aware of use a deterministic model. However, we are happy to add any additional evaluations that the reviewer suggests.
>
>
> ### Limitations
> See general discussion point (G4)
>
> ### Applications beyond Computer Vision
> We believe our framework is very generally applicable, and would be useful for a wide range of applications. However, we agree with the reviewer and will clearly mention the scope of our empirical result in the abstract and introduction of the paper. We address the generality of our approach in the general discussion point (G2), and discuss new experiments and potential applications.

---

> > ### Comment · Reviewer_bLEB · 2023-08-17
> >
> > First of all, I'd like to offer an apology to the authors. It looks like my original rating does not correspond to the rest of my review. The rating should be higher.
> >
> > I thank the authors for the extensive rebuttal and additional experiments. They are well received and strengthen the experimental validation.
> >
> > Question re NeuralField-LDM, while the authors have stated the approach is "radically different," I find the inputs and outputs are compatible -- in the sense that a comparison would be possible (the method proposed here could ignore any depth input data). Am I right, or what would make the comparison not worthwhile?

---

> > > ### Author Response · Authors · 2023-08-17
> > >
> > > We greatly appreciate you considering our rebuttal! NeuralField-LDM primarily learns an **unconditional** diffusion model and does not show any results of inferring a distribution of 3D scenes conditioned on one image. Their paper shows some results of conditioned synthesis, where the conditioning inputs are bird's eye view (BEV) segmentation maps that include significant information about the complete scenes. It is unclear whether this method can be extended to achieve the same input-output behavior as our paper. Unfortunately, their code is not publicly available, making any comparison practically very difficult.

---

> > > > ### Comment · Reviewer_bLEB · 2023-08-17
> > > >
> > > > Thanks for the response. I think it may be an issue of terminology but I see both methods being able to take as input a collection of images and then producing novel views of a 3D scene consistent with those images. I understand your method can sample multiple consistent 3D scenes but for evaluation (and applications) one could (and actually may want to) take a single 3D sample and produce multiple views from it. That should enable comparison, under some metrics, to an approach like NeuralField-LDM.
> > > >
> > > > In any case, I'm not suggesting this is necessary for me to recommend acceptance. Only of interest given recent developments in the space.

---

> > > > > ### Author Response · Authors · 2023-08-18
> > > > >
> > > > > Thanks for the fast response! We would like to clarify our understanding of the experiment you are proposing. On a high level, NeuralField-LDM has 2 stages: They first reconstruct a large dataset of 3D scenes via a *deterministic* scene auto-encoder. They then train an *unconditional* generative diffusion model on these reconstructed 3D scenes.
> > > > >
> > > > > We understand that you are proposing to include a comparison with stage 1, i.e. the deterministic scene auto-encoder. This ingests a set of images and maps them to a 3D scene that can be rendered from novel views. NeuralField-LDM uses stage 1 only during training to generate the 3D training dataset for stage 2.
> > > > >
> > > > > However, we note that this deterministic model *cannot* plausibly reconstruct parts of the 3D scene that were not observed in the input image. Any part of the 3D scene not seen will collapse to the mean estimate and hence be blurry. The input-output behavior here is identical to pixelNeRF, which we already compare with.
> > > > >
> > > > > The diffusion model in NeuralField-LDM, which generated all the main results in their paper, does not map an image to a sharp 3D scene since it is unconditional. Hence, we cannot compare with that part of the pipeline.
> > > > >
> > > > > We are happy to include a comparison with (our implementation of) the deterministic scene auto-encoder, which will lead to similar results as pixelNeRF (i.e., blurry reconstructions in parts of the scene that aren't visible in the input images). We could include this in Fig. 4 in the main paper.
> > > > >
> > > > > Please let us know if this sounds good!

---

### Official Review · Reviewer_5usH · 2023-07-07

**Soundness:** 3 good
**Presentation:** 4 excellent
**Contribution:** 3 good
**Rating:** 6
**Confidence:** 4

**Summary:**

This paper focuses on denoising diffusion models, a type of generative model used to capture complex signal distributions. However, current approaches can only model distributions when training samples are available, which is not always the case in real-world tasks. In fields like inverse graphics, where the goal is to sample from a distribution of 3D scenes based on a given image (without having access to ground-truth 3D scenes), this limitation poses a challenge.

To address this, the authors introduce a new class of denoising diffusion probabilistic models. These models learn to sample from distributions of signals that are never directly observed but instead measured through a known differentiable forward model. This forward model generates partial observations of the unknown signal. The authors integrate this forward model directly into the denoising process.

During testing, this approach enables sampling from the distribution of underlying signals consistent with a given partial observation. The authors demonstrate the effectiveness of their method on three challenging computer vision tasks. For instance, in inverse graphics, they show that their model allows direct sampling from the distribution of 3D scenes consistent with a single 2D input image.

**Strengths:**

This paper proposes a new type of conditional diffusion model for 3D scene generation without referring to the underlying 3D signal. This is an essential problem in the present era. The writing is super clear and I can easily follow. The experiments is sound and demonstrate the effectiveness of the proposed method. Moreover, I do appreciate the in-depth analysis and formulation of the "3D generation w/o 3D data" challenge.

**Weaknesses:**

Though this method gives a theoretical formulation of this problem, I believe more elaboration on some details and comparisons with some baselines are required to address my concerns. I have left my questions below.

**Questions:**

1. Though this method gives a theoretical formulation of this problem, to some extent I find it quite similar to the underlying idea of RenderDiffusion (CVPR 23'), which also trains a diffusion model which each denoising step output the denoised X0.  Though Renderdiffusion does not leverage the trgt-ctxt pair for training, the "learning 3D from 2D projection" spirit looks similar. I would like to hear the author's comments on this.

2. I wonder if your method share the same setting with "Generative novel view synthesis with 3d-aware diffusion models." which trains an independent diffusion model on a particular scene (like Hydrant and a single scene from RealEstate10k? This is not a limitation but since EG3D/RenderDiffusion like method could perform well on a single category, e.g., FFHQ/AFHQ/Shapenet, is your method also capable of achieving that performance, though they do unconditional generation and your method is a conditional one.

3. Since currently GT 3D poses are required, I wonder is this method robust to the noisy pose problem? Besides, since your incorporate the volume rendering as the "forward" process, is the synthesized scene perfect view consistent?

4. I do appreciate your experiment on RealEstate10K, and I wonder whether your method could be generalized to the "scene-level" 3D generation such as urban / BlockNeRF-like setting? If not, what is the challenge.

5. What are the limitations / failure cases of this method and could you share some insight behind it?

6. Regarding the GAN inversion experiment, I wonder whether your method has any intrinsic advantages over 3D GAN inversion, since you used StyleGAN2 in the experiment while this method is designed for 3D. Comparison with "E3DGE: Self-Supervised Geometry-Aware Encoder for Style-based 3D GAN Inversion, CVPR 23" is welcome since you both adopt encoder-based pipeline.

7. I see that your method achieves much better performance against SparseFusion. I wonder the popular SDS loss is still helpful in your framework and do you see it as a necessary component in the future 3D generation task? Thanks for your elaboration.

**Limitations:**

Though some concerns, overall I find this paper a sound submission and I hold my current rating towards accept. I am looking forward to the authors' elaboration in the rebuttal.

---

> ### Author Rebuttal · Authors · 2023-08-10
>
> We thank the reviewer for their comments. The reviewer nicely summarizes our paper and notes that the “writing is super clear”, we solve “an essential problem in the present era”, and that our experiments “demonstrate the effectiveness...”. We now address the points raised in the review.
>
> ### Related Work
> **RenderDiffusion** :  See general comment (G1)
>
> **Generative novel view synthesis with 3d-aware diffusion models (GeNVS)**: Please note that GeNVS is concurrent to our submission. Both our method and GeNVS train on a dataset of scenes, and not a single scene. However, there are fundamental differences. While we integrate the forward model with the denoising diffusion architecture to enable 3D scene generation, GeNVS is limited to only generating 2D images from the diffusion model. This is a crucial difference: Genvs is *not* a 3D generative model, i.e. it is not capable of generating 3D scenes. Rather, it can only generate *novel views* of 3D scenes, and thus would require score distillation to obtain a 3D scene, a limitation that GeNVS shares with SparseFusion, which we benchmark with.
>
> **EG3D**: EG3D is a 3D GAN that is trained from 2D images. It belongs to a class of 3D GANs that are primarily limited to unconditional modeling of simple objects (as the reviewer correctly notes). Our approach, for the first time, enables learning 3D diffusion models of complicated scenes, such as RealEstate10k - note that 3D GANs have never been able to demonstrate generative modeling of scenes of comparable complexity. We have not explored training on monocular image collections in our paper. Monocular images make the learning problem under-constrained (because of the lack of ctxt-trgt pairs), and exploring how our theoretical formulation can be extended to such cases is a very interesting problem that we leave for future exploration.
>
> ### Robustness to noisy poses
> We agree with the reviewer that our method requires GT 3D poses. We performed an experiment where we added noise to the pose parameters. The results gracefully degrade - training with 5% noise added to the camera translation parameters makes the results blurry, and leads to worse performance quantitatively, see Fig. 1 and Tab. 1 in the rebuttal pdf. Relying on GT poses is a very common requirement in contemporary 3D reconstruction methods, and we will mention this in our limitations. Removing this reliance on known accurate poses is an important problem for future research.
>
> ### View consistency
>
> Volume rendering can indeed lead to imperfect view consistency, depending on the quality of training supervision. As we train on videos where we have reasonably dense supervision, we found our results to be highly 3D consistent. To demonstrate this, we visualize point clouds extracted from the reconstructed 3D volume from extreme out-of-distribution viewpoints, see Fig. 5 in the rebuttal pdf. These point clouds demonstrate that our method extracts reasonable surfaces in 3D.
>
> ### Scene-level generation
> We thank the reviewer for their suggestion. We show results on trajectories that enable moving from one room to another in the case of RealEstate10k, and moving all around an object in the case of Co3D. Extending to city-scale trajectories would require tackling similar challenges as BlockNeRF, where the method would need to forget context images that are further in the past, and be able to merge several independent NeRFs. The main limitation at this point is the compute resources required to train and sample longer trajectories. We believe that these limitations are transitory and we would see large scale scene synthesis in the near future, with more optimized architectures and better hardware.
>
> ### Limitations
> See general comment (G4)
>
> ### 3D GAN inversion
> Our 2D GAN experiment showed that our framework is capable of using non-linear GAN-based forward models, and can help with inference from partial data. 3D GAN inversion is interesting as, similar to our inverse graphics task, it also involves tackling the uncertainties that arise from projection, while constraining the reconstructions to the GAN manifold. Our framework should provide the same advantages it currently does over the deterministic baselines on inverse graphics. Nevertheless, we aim to conduct this experiment for the revised version of our paper.
>
> ### SparseFusion and SDS
> Thank you for the insightful question! Our paper shows that SDS is not **necessary** for 3D generation, as we demonstrate that it is possible to learn a 3D generative model without SDS. This resolves some of the limitations that come with SDS: it has mode-seeking behavior, it requires costly test-time optimization, and it usually relies on monocular image priors that lead to 3D artifacts (like the infamous Janus artifacts). While some recent approaches make progress on some of these limitations, we show that we can completely side-step these issues using our novel approach that natively learns 3D models. At the same time, we also believe that SDS can be **complementary** to our approach: we could use SDS as an additional form of supervision at training time to supervise text conditioning, use it at test time to constrain the sampled scenes conditioned on text, or to extract high-fidelity mesh models. We will add a discussion of these exciting topics to our future work section.

---

> > ### Comment · Reviewer_5usH · 2023-08-13
> >
> > Thanks for the rebuttal of the author. Though this method has some limitations, the author's rebuttal has addressed most of my concerns. I think it worth an acceptance and presentation in the main nips conference.

---

### Author Rebuttal · Authors · 2023-08-09

We thank all reviewers for their efforts. The reviewers note that we solve  “an essential problem in the present era” (5usH), we tackle a “key challenge” in training 3D diffusion models (bLEB), our architecture is “well-designed and intuitive” (LcVC), and “results seem strong, both visually and quantitatively” (zgt6).

The reviewers have proposed additional experiments that will better highlight our work's strengths and limitations. **We are happy to report that we have been able to execute the majority of the experiments that the reviewers proposed, with favorable results, which we are excited to include in the paper.**. We now offer clarifications to some points shared across reviewers.

### (G1) Differences to RenderDiffusion
RenderDiffusion indeed shares some of the motivation of our approach, which is training 3D diffusion models from 2D data, which we discuss in our paper (L162-163). However, our approach differs from RenderDiffusion in several critical points, which uniquely enables our method to go beyond simple, single-object scenes:
1. RenderDiffusion learns an unconditional model.
2. RenderDiffusion requires canonical camera poses, i.e., all of the objects have to be oriented according to a canonical reference frame. This is a major limitation, as canonical poses do not exist for compositional real-world scenes. Our approach does not require canonical poses.
3. Through not requiring canonical camera poses, and with the proposed conditional diffusion that is not limited to monocular supervision, our model for the first time departs from simple, single-object scenes and succeeds at 3D generative modeling of room-scale, RealEstate10k scenes, a previously unachieved feat for 2D-to-3D generative models.

In addition, we provide a general theoretical formulation of conditional diffusion through forward models and define the conditions under which we can solve this problem optimally.  We also present empirical results for different applications.

### (G2) Generality

bLEB appreciates the generality of our formulation but notes the lack of applications outside of vision. Our theoretical formulation is general and not limited to forward models in computer vision. We demonstrate results on distinct, diverse, and important vision problems. We already mention the scope of our experimental results in the paper (lines 13-14). However, we agree that further clarification is beneficial. We will update the introduction and abstract to clarify that we only demonstrate applications in computer vision and will mention other domains only in the future work section.

zgt6 mentions the requirement on multiple observations. While this is true, and we demonstrate several practical applications, we further explored the nature of observations necessary for training. We add an inpainting experiment using multiple observations created from partially observed signals during training. This task has parallels in many domains, such as medical imaging or audio processing, where the goal is to complete missing information. We provide more details for this experiment in the response to zgt6 (“Multiple views required for training”), and Tab.1, Fig.4 of the response PDF.

More possible applications of our framework:
- Physics simulators can serve as forward models (see PAC-NeRF: Physics Augmented Continuum Neural Radiance Fields for Geometry-Agnostic System Identification [Li et al., 2023]) to learn the physical properties of objects by training on videos.
- Physically-based sound synthesis as forward models (see Singing Voice Synthesis Using Differentiable LPC and Glottal-Flow-Inspired Wavetables [Yu and Fazekas, 2023]) can be used to predict parameters of human vocal chords by training on singing voice datasets.

We believe that our framework would enable uncertainty-aware inference of many more physical systems.


### (G3) Evaluations


We evaluate against the state of the art in 3D reconstruction. While SparseFusion only uses the Co3D dataset for evaluation, we also use the very challenging scene-level  RealEstate10k dataset. We show results on more complex settings compared to the SparseFusion paper.  We only use a single input image (unlike >=2 for SparseFusion), and do not assume GT object masks, instead modeling the whole scene including background. This makes our task significantly more challenging. Thus, in our paper, we only trained on hydrants (a concurrent work, GeNVS, also only trained on hydrants).

We agree with LcVC that training a general model on 10 Co3D categories will strengthen the paper. In the reponse PDF (Tab. 1, Fig. 2), we show the results of such a general model, and compare to pixelNeRF and SparseFusion, significantly outperforming them. Sparsefusion fails to generate any reasonable results in this multi-class setting. Note that the SparseFusion paper only trained category-specific models. **To the best of our knowledge, these are the first-ever results of category-agnostic diffusion models on Co3D.**

We further show preliminary results on training on renderings of 15k objects from the large-scale object-centric Objaverse dataset, see Fig. 3 of the response PDF.

These results demonstrate that our method is capable of modeling complex distributions, and that it strongly outperforms all existing baselines.

### (G4) Limitations
Reviewers asked about the limitations of our approach. They are mentioned in our supplemental pdf, see Section 4 (L191-202). Among other points, we note the complexity of training and inference. We will move the limitations to the main pdf in the revised version.

### (G5) Updated Figure
While most reviewers appreciate our exposition, zgt6 mentions that some application descriptions are unclear. We have updated the overview figure of our main application of inverse graphics that we hope better explains the main application, see Fig. 6 of the response PDF.

---

### Decision · Program_Chairs · 2023-09-21

**Decision:**

Accept (spotlight)

**Comment:**

The reviewers were generally positive on the paper.  It proposes a novel approach to a challenging and interesting problem with solid results.  The paper is clearly above the bar and should be accepted.  I encourage the authors to further revise the manuscript based on the feedback from the review and discussion process for the final copy.